# A macrophage-predominant immunosuppressive microenvironment and therapeutic vulnerabilities in advanced salivary gland cancer

Erika Zuljan[1], Benjamin von der Emde[2], Iris Piwonski[3], Ana Pestana[4,5], Konrad Klinghammer[2,5,6], Andreas Mock[7,8,9,10], Peter Horak[7,9,10], Christoph Heining[11,12,13], Frederick Klauschen[3,5,8,14], Ina Pretzell[15,16], Melanie Boerries[17,18], Christian H. Brandts[19,20], Simon Kreutzfeldt[7,9,10], Maria-Veronica Teleanu[7,9,10], Daniel Hübschmann[7,9,10], Luc G. T. Morris[21], Max Heiland[6,22], Ulrich Keller[2,5,6], Thomas Conrad[23], Hanno Glimm[11,12,13], Stefan Fröhling[7,9,10,24], Sebastian Ochsenreither[2,4,5,6], Ulrich Keilholz[4,5,6], Eric Blanc[1,25], Dieter Beule[1,25] & Damian T. Rieke[2,4,5,6,25] ✉

Salivary gland cancers are rare, diverse malignancies characterized by poor response to immunotherapy. The tumor immune environment in these cancers remains poorly understood. To address this, we perform an integrative analysis of the tumor immune microenvironment in a large cohort of advanced salivary gland cancer samples. Most tumors exhibit low immune activity with limited immune cell infiltration. Inflammation is linked to higher tumor mutational burden in non-adenoid cystic carcinoma histologies. Subtype specific expression of immune checkpoints is identified with prominent expression of *VTCN1* in luminal-like cells within adenoid cystic carcinoma. Macrophages with immunosuppressive properties dominate the immune microenvironment across subtypes. Responses to immunotherapy are limited and associated with a higher ratio of T-cells relative to macrophages in individual cases, warranting further investigation. Here, we show an immunosuppressive environment in salivary gland cancers and identify subtype-specific immune vulnerabilities that could inform tailored therapeutic strategies.

Salivary gland cancers (SGC) are a rare and heterogeneous group of malignancies that arise from major and minor salivary glands. SGC account for 5% of all head and neck cancers and comprise more than 20 different histologies[1]. The prognosis is poor in the recurrent and metastatic setting, with a median OS of 15 months after appearance of distant metastasis[2]. No approved therapies for advanced SGC exist. Histological subtypes are often not adequately represented in available clinical trials with the exception of adenoid cystic carcinoma (ACC). ACC is among the most common malignant subtypes, defined by a recurrent *MYB-NFIB* gene fusion and often presents with slow tumor growth and poor response to systemic therapy[3]. Yet, different molecular and clinical subsets of ACC have been described[4]. In contrast, non-ACC tumors typically show an aggressive clinical course and often harbor targetable molecular alterations[5].

**Fig. 1 | Study cohort and data. A** Clinical characteristics of the cohort, including tumor entities, therapy status (therapy prior to sequencing), site of biopsy, age and sex are presented. **B** Availability of sequencing data and number of samples (do not equal to number of patients on the left), as well as overlapping data availability, is provided. RNA-seq data were evaluable for a majority of patients (*n* = 93, n_samples 95). (ACC Adenoid Cystic Carcinoma, ADC Adenocarcinoma, BCC Basal Cell Carcinoma, MEC Mucoepidermoid Carcinoma, SDC Salivary Duct Carcinoma, LSG Large Salivary Glands, WGS Whole Genome Sequencing, WES Whole Exome Sequencing. Figure created using the Mind the Graph platform (www.mindthegraph.com).

Immune checkpoint inhibition yielded low response rates in clinical trials in advanced salivary gland cancer. In a study of pembrolizumab in PD-L1 positive salivary gland cancer, 26 patients were enrolled and three objective responses (all PR) could be reported in adenocarcinoma (*n* = 2) and high-grade serous carcinoma (*n* = 1)[6]. Similarly, in another phase 2 trial of pembrolizumab, in 109 patients with pretreated SGC, including few ACC, an objective response rate of 4.6% was observed[7]. In this trial, a higher response rate (10.7%) was noted in PD-L1 positive disease. A comparable response rate of 4.2% (salivary duct carcinoma) was reported in a retrospective multicenter analysis of nivolumab in 24 patients with SGC[8]. The combination of nivolumab and ipilimumab was tested in separate cohorts of 32 patients each with ACC and non-ACC[9]. In this trial, the primary efficacy endpoint of at least 4 objective responses was met in the non-ACC (ORR 16%) but not in the ACC cohort (ORR 6%). Similarly, no objective responses were noted in 20 ACC patients randomized 1:1 to pembrolizumab with or without radiotherapy[10]. These data show an overall limited efficacy of immune checkpoint inhibitors in advanced SGC[6–11]. Among these prospective trials, no clear subtypes and predictive biomarker for the use of immune checkpoint inhibitors could be identified, although condensed data suggest low activity especially in ACC. The use of immunotherapy outside of clinical trials is therefore not routinely recommended in SGC and predictive biomarkers and suitable treatment strategies are urgently required[12]. Available data on the tumor immune microenvironment in SGC show subgroup-specific differences with an immune-excluded microenvironment in ACC, but only limited data are available in the recurrent and/or metastatic setting[13–15]. Here, we provide a multi-omics analysis of the tumor immune microenvironment in a cohort of recurrent and/or metastatic salivary gland cancers, thus representing a potential intention-to-treat cohort for systemic therapies. Taken together, these results provide a comprehensive insight into the tumor immune microenvironment in advanced SGC.

## Results

### Cohort characteristics

A total of 104 patients with recurrent/metastatic salivary gland cancer from the MASTER program were analyzed. The most common tumor entity was ACC (58%, 61/104 patients). In addition to ACC, a further 12 tumor entities were included. The median age at the time of tumor biopsy was 46 years (IQR 41–61), and 46% of patients were female. One patient was younger than 18 years at the time of diagnosis. The most common primary tumor site was the parotid gland in 47% of all samples. Two-thirds of patients had received prior systemic therapy before inclusion in the MASTER program, with a median of one line (IQR 0–2) of therapy. Median duration of prior therapy lines was 139 days (IQR 57.5–153). A summary of clinical data is provided in Fig. 1, Table 1 and Supplementary Data 1.

Bulk RNA-seq data was available for 93 patients (95 samples). Whole-Exome (WES) and Whole-Genome sequencing (WGS) data was available for 55 and 50 patients (52 samples), respectively. Single-nuclei transcriptome sequencing was performed in selected samples (*n* = 13), representing inflammation-high and inflammation-low in ACC and non-ACC histologies. A summary of molecular data layers is provided in Fig. 1 and Supplementary Figure 1. Immunohistochemical analyses were performed in 44 samples (17 of which were part of the MASTER cohort and 15 with available bulk RNAseq data).

*MYB-NFIB* fusions were the most common genetic alteration, identified in 60% of ACC samples (Fig. 2A). An additional 6% of ACC harbored *MYBL1-NFIB* fusions. These proportions are supported in the literature[16]. None of the non-ACC samples had a *MYB-NFIB* fusion. A *PLAG1*-fusion suggesting carcinoma ex pleomorphic adenoma was identified in one sample previously diagnosed as ACC. Two adenocarcinoma NOS showed specific alterations: one sample had an *EWSR1*-fusion suggesting clear-cell carcinoma, and another had a focal amplification of the *HER2* gene (CN > 10) suggesting salivary duct carcinoma (SDC). *TP53* was the most commonly altered gene in non-ACC (26%), followed by *PTEN* (12%) (Fig. 2B). Most common short variants in ACC affected *NOTCH1* (18%), *BCOR* (13%), *KDM6A* (10%), *ARID1A* (13%), and *ACTB* (13%) (Fig. 2B). Among CNVs the most common alteration was the deletion of the Cyclin Dependent Kinase Inhibitor 2 (*CDKN2A, CDKN2B, CDKNC* total: 8%). 3 out of 7 SDCs had a focal amplification of *HER2*. Overall the chromosomal aberration index (portion of DNA affected by CNVs) was significantly higher in non-ACC (*Wilcoxon*, *p* < 0.0001, Supplementary Fig. 2A). TMB was significantly higher in non-ACC than ACC samples (*Wilcoxon test*, *p* = 2.9e−06***, Fig. 2C). The TMB of ACC was also in the lower spectrum when compared to other TCGA entities (median: 0.8 Mut/Mb, 8/33, Supplementary Fig. 2B). Mutational signatures were extracted for WGS and WES data (Supplementary Fig. 2C). Recurrent signatures included

**Table 1 | Baseline clinical characteristics of the cohort. The table shows clinical data of the cohort stratified by tumor entity (ACC vs non-ACC). Clinical data includes sex, age at biopsy, primary site, tumor entity, and information on prior therapies. Due to rounding percentages may not add up to 100. Detailed, per-patient data can be found in Supplementary Table 1**

| Summary of the clinical data | ACC (n = 60) | non-ACC (n = 44) | all (n = 104) |
|---|---|---|---|
| Sex | | | |
| female | 38 (63%) | 10 (23%) | 48 (46%) |
| male | 22 (37%) | 34 (77%) | 56 (54%) |
| Age at date of biopsy | | | |
| 0–30 | 5 (8%) | 3 (7%) | 8 (8%) |
| 31–50 | 27 (45%) | 14 (33%) | 41 (40%) |
| 51–70 | 26 (43%) | 25 (58%) | 51 (50%) |
| >70 | 2 (3%) | 1 (2%) | 3 (3%) |
| NA | 0 (0%) | 1 (2%) | 1 (1%) |
| Tumor site | | | |
| Parotid gland | 21 (35%) | 27 (63%) | 48 (47%) |
| Submandibular gland | 9 (15%) | 12 (28%) | 21 (20%) |
| nasopharynx NOS | 6 (10%) | 0 (0%) | 6 (6%) |
| large salivary glands NOS | 2 (3%) | 1 (2%) | 3 (3%) |
| other | 22 (37%) | 3 (7%) | 25 (24%) |
| NA | 0 (0%) | 1 (2%) | 1 (1%) |
| Entity | | | |
| ACC | 60 (100%) | 0 (0%) | 60 (58%) |
| Adenocarcinoma NOS | 0 (0%) | 8 (18%) | 8 (8%) |
| Mucoepidermoid carcinoma | 0 (0%) | 7 (16%) | 7 (7%) |
| Basal cell carcinoma | 0 (0%) | 6 (14%) | 6 (6%) |
| Salivary duct carcinoma | 0 (0%) | 6 (14%) | 6 (6%) |
| Malignant mesenchymal tumor | 0 (0%) | 5 (11%) | 5 (5%) |
| Carcinoma ex pleomorphic adenoma | 0 (0%) | 4 (9%) | 4 (4%) |
| Acinic cell carcinoma | 0 (0%) | 3 (7%) | 3 (3%) |
| other | 0 (0%) | 5 (11%) | 5 (5% |
| Prior therapy lines* | | | |
| Chemotherapy | 26 (43%) | 26 (59%) | 52 (50%) |
| TK-inhibition | 8 (13%) | 3 (7%) | 11 (11%) |
| Immune checkpoint inhibition | 3 (5%) | 6 (14%) | 9 (9%) |
| HER2-blockade | 0 (0%) | 6 (14%) | 6 (6%) |
| EGFR-inhibition | 1 (2%) | 5 (11%) | 6 (6%) |
| Number of previous therapy lines | | | |
| 0 | 23 (38%) | 13 (30%) | 36 (35%) |
| 1 | 22 (37%) | 14 (32%) | 36 (35%) |
| 2 | 10 (17%) | 7 (16%) | 17 (16%) |
| 3 | 3 (5%) | 3 (7%) | 6 (6%) |
| >3 | 2 (3%) | 7 (16%) | 9 (9%) |

clock-like signatures (SBS1,5), damage by reactive oxygen species (SBS18), platinum-chemotherapy-related signature (SBS31, prevalent in patients with prior treatment) and APOBEC-mutagenesis-signatures (SBS2,13, 6 samples including 3 mucoepidermoid carcinomas, Supplementary Fig. 2D). A summary of molecular tumor profiles is provided in Fig. 2A. Fusion status of each patient is provided in Supplementary Data 1.

We analyzed bulk RNA-seq data to identify gene expression differences. Principal component analysis revealed most variance (13%) to be explained by tumor entity ($p = 4.5e-10$***, Supplementary Fig. 3A, D). A small portion of variance was explained by the biopsy site (~4%), which was associated with PC3 and PC5. (Supplementary Fig. 3D). PC1, which also separated ACC from other entities, was related to immune response (Supplementary Fig. 3B, C). Differentially expressed genes between ACC and non-ACC were also related to inflammation (AUC = 0.7, p.adj = $5.2e-8$***), cell cycle (AUC = 0.8, p.adj = $2.6e-10$***) and cell cycle in T-cells (AUC = 0.8, p.adj = $2.4e-4$***) with lower expression in ACC (Supplementary Fig. 3E).

## Salivary gland cancers cluster into 3 groups of immune infiltration

To further analyze tumor inflammation in advanced SGC, bulk gene expression data was analyzed. Six different published functional inflammation signatures were used to identify samples with an inflamed tumor immune microenvironment (TIM) (Fig. 3A). Additional measures of immune infiltration besides the 6 GSVA-based signatures shown in Fig. 3A were computed. All of the mentioned immune scores showed significant correlation between each other after multiple testing corrections (Supplementary Fig. 4A).

Hierarchical clustering based on GSVA scores identified 3 clusters of immune-infiltration (Fig. 3A) with 35% of samples belonging to the immune-high cluster. We selected three clusters instead of two since the medium cluster demonstrated significant differences in inflammation compared to the high and low clusters, as illustrated in Fig. 3A, C and Supplementary Fig. 4B. ACC were significantly underrepresented in the immune-high cluster (Fisher test, p = 0.002**, OR = 0.3). We could test inflammation differences in a limited number of histologies due to the low sample size per group. Both mucoepidermoid carcinoma (MEC, n = 7) and SDC (n = 7) had a similar level of inflammation which was significantly higher than in ACC (n = 57) (Supplementary Fig. 4C). In order to validate these findings in a larger cohort, we performed an integrated analysis with previously published SGC datasets[9,13] which further confirmed 3 clusters of immune infiltration with ACC being significantly underrepresented in the immune-high group (Fig. 3B). Importantly, a small immune-high ACC subgroup could be validated in all 3 cohorts (Fig. 3B). After data integration, a sufficient number of samples for histology-specific analyses (n > 20 each) was evaluable for ACC, myoepithelial carcinoma and SDC and revealed significant differences between the subtypes (Supplementary Fig. 4D). Myoepithelial carcinoma (n = 26) had overall lower inflammation than ACC (n = 158) and SDC was confirmed as being the highest inflamed entity (n = 46). Compared to pan-cancer TCGA data, inflammation (IFNG score) in the ACC cohort was comparable to those of the 5 least inflamed TCGA cohorts (Supplementary Fig. 4I). Other entities showed a much higher variance in their inflammation score, however inflammation scores of immune-high samples aligned with the 5 highest inflamed TCGA cohorts (Fig. 3C).

Immune clusters were validated immuno-histochemically in a tissue microarray with available bulk sequencing data (n = 14 evaluable, Fig. 3D and Supplementary Fig. 4E–H). CD3 staining intensity was significantly correlated with the IFNG score (Spearman, rho = 0.74, p = 0.002**) (Fig. 3D).

## Correlates of tumor inflammation in advanced SGC

We tested for the association of different parameters with the immune clusters and inflammation scores. ACC-histology (One-way anova, p.adj = 0.003 **) and closely related cell-of-origin-group (One-way anova test, p.adj = 0.0003 ***) as well as tumor purity (One-way anova test, p.adj = 0.0003***) and TMB (One-way anova test, p.adj.=0.04*) were significantly associated with inflammation (Fig. 3E). No significant

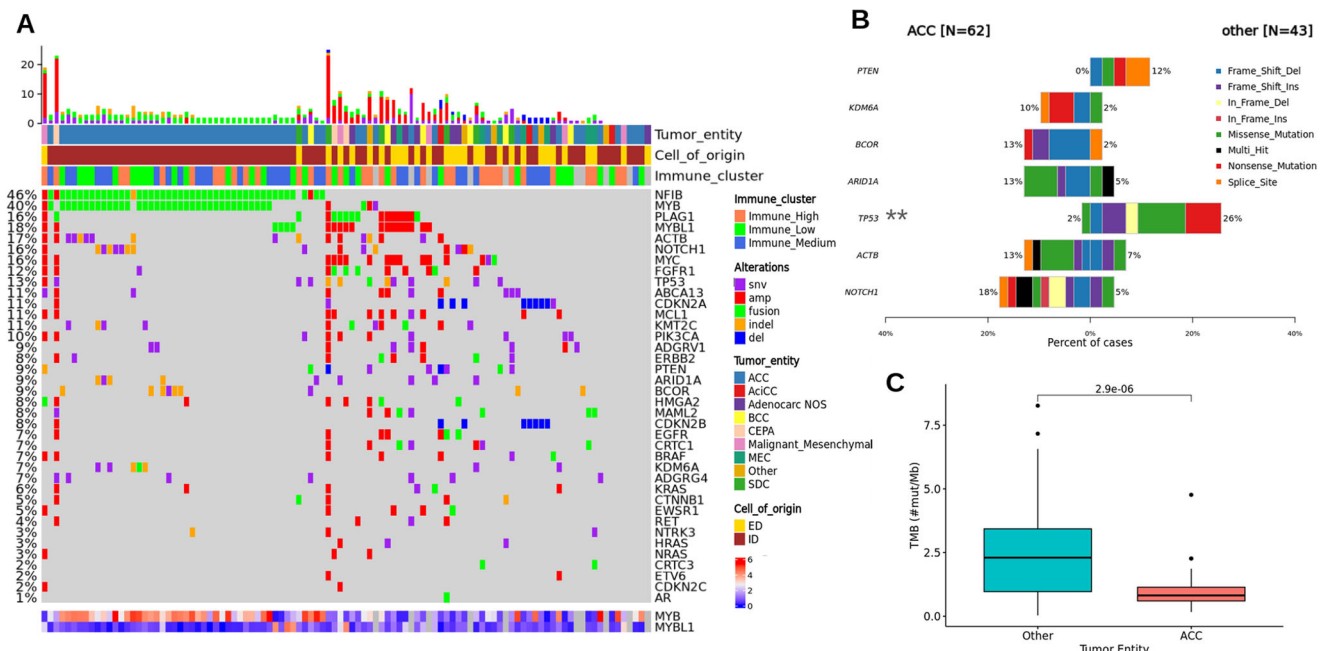

**Fig. 2 | Summary of molecular alterations. A** Oncoprint with molecular alterations in a set of selected genes commonly affected in SGC ($n = 104$). On the left, the percentage of samples with one or more alterations in each listed gene is shown. The top bar shows the total number of alterations in the specific sample. The heatmap at the bottom represents the expression of *MYB* and *MYBL1* genes.

**B** Co-barplot with most common SNVs in ACC and non-ACC. *TP53* alterations are the only SNVs significantly enriched in non-ACC. **C** Tumor mutational burden was significantly higher in non-ACC ($n = 43$, median = 2.3, iqr = 2.5, max = 8.3, min = 0.03) compared to ACC samples ($n = 60$, median = 0.82, iqr = 0.54, max = 4.7, min = 0.17) (Wilcoxon test, two-sided).

associations were found for sample origin (metastatic/primary), primary site (large salivary glands/other), age, cohort batch, sex, and therapy status for several therapies (received therapy/therapy-naive) after multiple testing correction (Fig. 3E).

TMB and IFNG score showed a weak positive correlation (Spearman, rho = 0.31, p = 0.002**, Supplementary Fig. 5) in the combined WES and WGS data ($n = 103$). This correlation was mostly driven by non-ACC samples (Supplementary Fig. 5A, B). As expected, inflammation was significantly negatively correlated with tumor purity (computed based on WGS/WES data) in both ACC and non-ACC (Spearman, rho = 0.42, p = 2.4e-5***, Fig. 3E, Supplementary Fig. 5C).

Non-synonymous mutations in immunotherapy-relevant genes[17] were slightly enriched in ACC compared to non-ACC (Fisher test, p = 0.006**, OR = 1.6), however no positive selection could be identified in these genes (no difference in abundance of synonymous and nonsynonymous mutations) (Supplementary Fig. 5D). Among mutational signatures, clock-like signature 1 was more prevalent in the immune-low subgroup but impacted by ACC status, TMB and WGS sequencing. An APOBEC-mutational signature was associated with higher antigen processing machinery (APM) in few evaluable samples (Supplementary Fig. 2C, Supplementary Fig. 5E–G).

ACC comprises clinically distinct subgroups. We therefore tested potential mediators of inflammation in ACC samples separately (Supplementary Fig. 6). A marginally significant impact of metastatic site (lung) compared to primary tumor on tumor inflammation was observed in ACC samples only (Supplementary Fig. 6A, B). The ACC score, used to discriminate ACC subgroups 1 and 2 was negatively correlated with inflammation (Pearson, rho = −0.3, p = 0.03*, Supplementary Fig. 6B). A significant association with the APM was identified in these samples (Pearson, rho = −0.49, p = 0.0002***, Supplementary Fig. 6C). The negative correlation suggests a deficient antigen processing machinery mediating immune exclusion in ACC1. As expected, ACC1 samples were found to have a significantly worse prognosis than ACC2 (Supplementary Fig. 6D).

## TIM composition analysis reveals macrophage-dominance in SGC

To identify the TIM cell composition in advanced SGC, single nuclei RNA-seq data was analyzed for 13 samples. The single nuclei cohort comprised 5 ACC and 8 non-ACC (2 basal cell carcinoma, 1 carcinoma ex pleomorphic adenoma, 2 adenocarcinoma NOS, 1 carcinosarcoma, 1 mucoepidermoid carcinoma, 1 salivary duct carcinoma). After quality control and filtering the total amount of cells was 85,142.

Table 2 shows the sample origin, tumor entity and quality metrics for each sample sequenced. Data analysis revealed 23 different cell clusters, which could be assigned to 6 major cell types: fibroblasts, endothelial cells, immune cells, alveolar cells (present only in the lung metastases) and malignant cells (Fig. 4A, B and Supplementary Fig. 7A–N). More than half of all identified immune cells came from 2 lymph node metastases (Fig. 4A, P-18 and P-66). Therefore, these samples were removed to further analyze TIM composition. The majority of immune cells were labeled as macrophages (*mean: 60% sd: 25*), followed by T-cells (*mean: 22%, sd: 18*), dendritic cells (*mean: 6%, sd: 5*) and plasma cells (*mean: 4% sd: 5*) (Fig. 4C–E). The least abundant immune cells were B-cells, which were almost only present in lymph node metastases. Samples labeled as immune high in bulk data analysis had more immune cells than those labeled as immune low or medium (Fig. 4A). Bulk based scores, in particular IFNG score were also in agreement with immune cell proportions from single nuclei data (Spearman, rho = 0.8, *p = 0.002**, Supplementary Fig. 7O). Macrophages and T-cells were analyzed in depth (Supplementary Figs. 8 and 9) after exclusion of one sample (P-31), because of remarkably high macrophage infiltration, resulting from the phagocytosis of necrotic tissue, as confirmed by microscopy. Bulk data analysis revealed a predominance of M2-polarized macrophages as compared to M1 or M0 macrophages (Supplementary Fig. 8A). Single-cell analysis and immunohistochemistry confirmed these findings with a majority of macrophages exhibiting high expression of genes associated with an M2-polarized or tumor-associated macrophage (TAM) phenotype (Supplementary Fig. 8C, G, H, J). M2 markers including *MSR1, MRC1,*

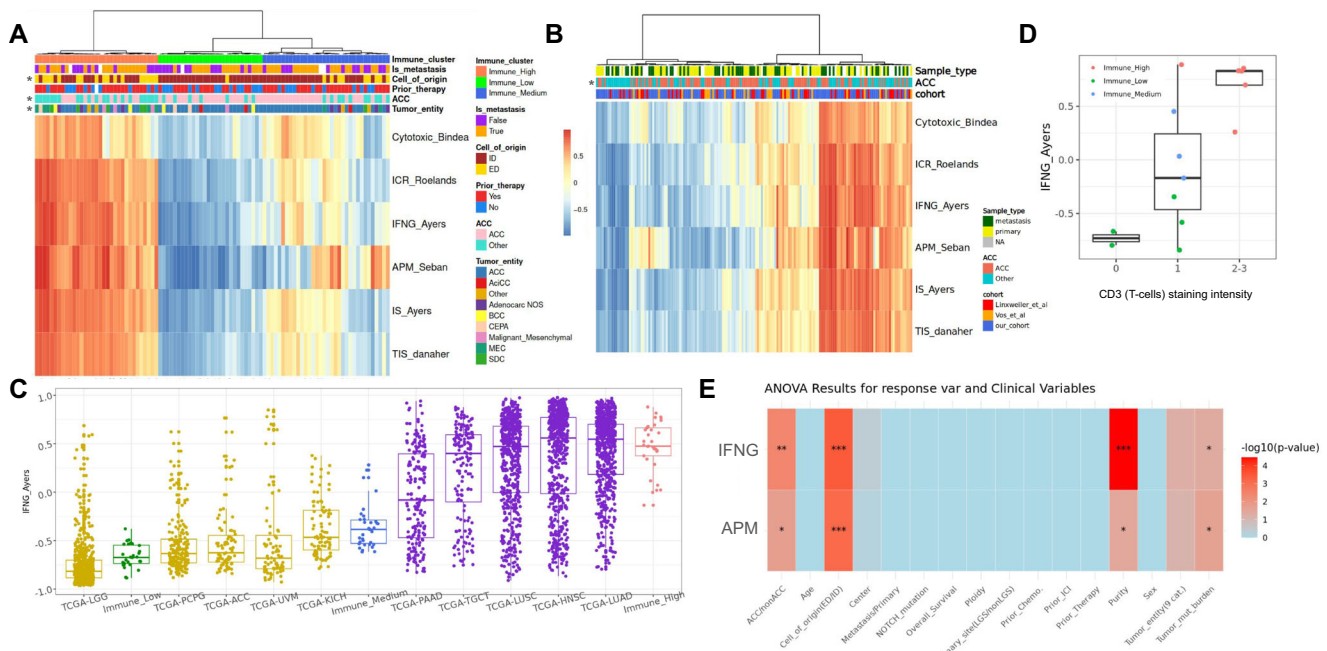

**Fig. 3 | Advanced SGC cluster into 3 groups of immune infiltration. A** Heatmap of GSVA scores of 6 immune signatures (see methods) for all samples (n = 95). Samples were clustered by hierarchical clustering and annotated by tumor entity (ACC/non-ACC or 9 categories of tumor entities), cell-of-origin group (ID/ED), sample type (metastasis/primary) and prior systemic therapy status. **B** GSVA scores of 6 immune signatures were calculated in an integrated analysis together with previously published cohorts (n = 198). Samples were annotated by tumor entity (ACC/non-ACC), sample type (metastasis/primary) and cohort (MASTER cohort/ Linxweiler et al/Vos et al). **C** Comparison of immune clusters (Immune-high n = 33 median = 0.48, iqr = 0.29, max = 0.88, min = −0.13; Immune-medium n = 34 median = −0.38, iqr = 0.24, max = 0.28, min = −0.62; Immune-low n = 28 median = −0.67, iqr=0.19, max = −0.38, min = −0.89) to TCGA most (PAAD n = 183 median = −0.08, iqr = 0.87, max = 0.94, min = −0.83; TGCT n = 156 median = 0.40, iqr = 0.69, max = 0.88, min = −0.76; LUSC n = 553 median = 0.47, iqr = 0.68, max = 0.97, min = −0.91;

HNSC n = 566 median = 0.56, iqr = 0.79, max = 0.98, min = −0.87; LUAD n = 600 median = 0.55, iqr = 0.52, max = 0.97, min = −0.80) and least inflamed cohorts (LGG n = 534 median = −0.82, iqr = 0.18, max = 0.69, min = −0.96; PCPG n = 187 median = −0.63, iqr = 0.24, max = 0.62, min = −0.88; ACC n = 79 median = −0.63, iqr = 0.28, max = 0.77, min = −0.84; UVM n = 80 median = −0.68, iqr = 0.34, max = 0.85, min = −0.93; KICH n = 91 median = −0.47, iqr = 0.41, max = 0.38, min = −0.79). **D** Intensity of pan T-cell marker (CD3) versus IFNG bulk score. Samples were colored by immune clusters (n = 14, Immune-high n = 6 median=0.83, iqr=0.13, max=0.85, min=0.26; Immune-medium n = 3 median = −0.17, iqr=0.71, max=0.89, min=−0.84; Immune-low n = 5 median = −0.73, iqr=0.06, max = −0.67, min = −0.80). **E** One-way anova test was performed to analyze the association of several clinical parameters with immune scores (APM and IFNG). Corrected and log-transformed p-values are provided in the heatmap, showing a significant impact of tumor entity, cell-of-origin, tumor purity, and TMB.

*CD163*, and *CD68* were highly expressed in single cell data (Supplementary Fig. 8C, H). Highest M2 score was observed in macrophages derived from an ACC1 sample, however no clear differences could be recognized between ACC and non-ACC due to low sample sizes (P-33, Supplementary Fig. 8I). Principal Component Analysis (PCA) revealed that PC1 accounted for 10% of the variance and was linked to alveolar macrophage markers, such as *PPARG* (Supplementary Fig. 8D). These macrophages were exclusively found in samples from lung metastases, confirming their lung-specific origin. PC2 reflected a functional spectrum of TAMs in salivary gland cancers, showing a positive correlation (Pearson, rho = 0.5, p < 0.001***) with M2 marker expression (Supplementary Fig. 8G). The dominant gene in PC2, *F13A1*, has been previously described to promote fibrin cross-linking, creating a scaffold that facilitates cancer cell invasion and metastasis (Supplementary Fig. 8B, E)[18]. Conversely, genes associated with low PC2 values did not correlate with classical M1 markers but may represent a distinct population of immune-active or alternative-state macrophages (Supplementary Fig. 8F). Among the genes associated with low PC2 values 2 types of integrins were found (*ITGA4* and *ITGAX*, Supplementary Fig. 8B).

We additionally analyzed lymphocyte compositions. In bulk data a median of 49% of all T-cells were predicted to be CD4+ T helper cells, followed by 47% CD8+ cytotoxic cells (T-cells and NK-cells) and 4% T regulatory cells (Supplementary Fig. 9A). These data could also be validated in single-cell analysis (n = 11) and immunohistochemistry (IHC) (n = 43) (Supplementary Fig. 9B-K). In single-cell data and IHC

the CD8+ were slightly more abundant than the CD4+ cells (47% vs 41% in single-cell data) (Supplementary Fig. 9B, C). *TOX*, a transcription factor that regulates T-cell exhaustion[19] was highly expressed in Cytotoxic and T-regulatory cells in all samples (Supplementary Fig. 9I). The elevated expression of *TOX* in these T-cell subpopulations suggests a potential dysfunctional or exhausted phenotype.

We performed a deconvolution analysis (CIBERSORT, which showed the best agreement with single-cell based results and IHC) on bulk data to explore the TIM in a larger number of samples. Sixty-one samples had evaluable results from CIBERSORT (empirical p-value <=0.05) and could be used for subsequent analyses. Similar to single nuclei based results, most of the analyzed samples had a myeloid-dominant microenvironment (Fig. 5A). On average, myeloid cells made up 55% (41% Macrophages) and T-cells 25% of the TIM in the entire cohort (Fig. 5A). The relative proportion of T-cells, in particular CD8+ T-cells was slightly higher in the immune-high group, compared to the other two groups (Wilcoxon test, p = 1.7e-4***)(Fig. 5B). On the other hand, the relative proportion of M2 macrophages was lower in the immune high group compared to the immune low, however not significant (Wilcoxon test, p = 0.053) (Supplementary Fig. 10A). The proportions of the other immune cell types did not cluster by immune assignment nor by tumor entity (Fig. 5A). Multiple ANOVA tests did not identify significant associations of T-cell to macrophage ratio with clinical parameters such as tumor entity (ACC vs non-ACC), prior immune checkpoint inhibition (ICI) and sample type (metastatic vs

**Table 2 | Sample characteristics for single-nuclei sequencing.** The table shows clinical characteristics and quality metrics for the single-nuclei cohort. Clinical characteristics include site of biopsy, tumor entity, and inflammation group. Quality metrics are the number of nuclei per sample, the median UMI counts per nucleus, and the median number of transcripts per nucleus. (ACC=Adenoid Cystic Carcinoma, AdC=Adenocarcinoma, BCC=Basal Cell Carcinoma, CS=Carcinosarcoma, CEPA=Carcinoma ex Pleomorphic Adenoma, MEC=Mucoepidermoid Carcinoma, SDC=Salivary Duct Carcinoma)

| Sample | Site Biopsy | Entity | # nuclei | Immune cluster | median UMI counts/ nucleus | median transcripts/ nucleus |
|--------|-------------|--------|----------|----------------|----------------------------|-----------------------------|
| P-6 | lung | CS | 6640 | High | 4254 | 2235 |
| P-14 | lung | AdC | 2774 | High | 1740 | 1108 |
| P-18 | lymph node | SDC | 4344 | High | 5624 | 2277 |
| P-31_1 | primary | CEPA | 3957 | NA | 5034 | 2427 |
| P-31_2 | primary | CEPA | 3839 | NA | 4824 | 2395 |
| P-32 | primary | ACC | 4667 | Low | 2495 | 1545 |
| P-33 | skin | ACC | 5455 | Low | 3867 | 2064 |
| P-42 | lung | ACC | 10273 | High | 3750 | 1892 |
| P-46_1 | lung | BCC | 8082 | Low | 7149 | 2834 |
| P-46_2 | lung | BCC | 7864 | Low | 7152 | 2773 |
| P-66_1 | lymph node | BCC | 5973 | High | 4756 | 2292 |
| P-66_2 | lymph node | BCC | 4664 | High | 3961 | 2049 |
| P-74 | primary | MEC | 4802 | Medium | 2179 | 1358 |
| P-79 | skin | AdC | 2861 | Medium | 3261 | 1811 |
| P-85 | lung | ACC | 6497 | High | 2920 | 1804 |
| P-103 | lung | ACC | 4343 | Medium | 4095 | 2277 |

primary). When combining our data with 2 other published cohorts ($n = 101$) we did also not identify a difference in T-cell to Macrophage ratio in ACC vs non-ACC and in metastatic vs non-metastatic sites but found a significantly higher M2 proportion in ACC compared to non-ACC samples. (Wilcoxon test $p = 6e-05$***) (Fig. 5C, D). We stratified the entities further into ACC ($n = 45$), myoepithelial carcinoma ($n = 9$), and SDC ($n = 20$) and we still could not observe major differences in T-cell to Macrophage ratio, but still the highest M2 proportion in ACC (Supplementary Fig. 11A, B).

We additionally compared the T-cell to Macrophage ratio of this cohort and another advanced SGC cohort to several TCGA cohorts and healthy salivary glands. T-cell to Macrophage ratio in SGC was among the lowest, with a median T-cell/Macrophage ratio of both analyzed SGC cohorts below 1 (Supplementary Fig. 11C). Also the M2/ total Macrophage ratio was the highest in advanced SGC compared to other TCGA cohorts (Supplementary Fig. 11D). Both healthy and diseased SG had a low immune infiltration in the context of other TCGA cohorts (Supplementary Fig. 11E). A mean T-cell/Macrophage ratio of below 1 could also be validated immuno-histochemically (Fig. 5E). Our cohort analyses, spanning bulk RNA sequencing ($n = 61$), single-cell RNA sequencing ($n = 11$), and IHC ($n = 40$), consistently demonstrate a low T/M ratio, with mean values of 0.65, 0.73, and 0.8, respectively (Fig. 5E). This consistency across different methodologies suggests that this finding is a robust characteristic of advanced SGC.

Immunohistochemical analyses showed that macrophages tend to arrange themselves mostly in clusters at the tumor invasion front. T lymphocytes (here cytotoxic T lymphocytes) were only found sporadically in the majority of samples (Fig. 5F–I).

A trend to better overall survival was observed in samples with a high T-cell content, no impact of macrophages on survival was observed (Fig. 5J and Supplementary Fig. 10B).

In order to rule out the presence of muciphages as a cause of macrophage predominance we analyzed 34 samples via PAS staining (Supplementary Fig. 10C, D). Only one single PAS+ macrophage was identified in one sample. Furthermore we compared the TIM in SGC to the immune microenvironment in healthy salivary glands. We analyzed published salivary gland bulk ($n = 33$)[20–22] and single-cell data ($n = 2$)[22] (Supplementary Fig. 12). In contrast to SGC, B-cells and plasma cells constituted a large portion of immune cells whereas myeloid cells including macrophages only made up approximately 8% of all immune cells (Supplementary Fig. 12A–C). Deconvolution analyses predicted most macrophages in healthy SG to be M2 polarized, however single cell data showed different results. Only a few cells expressed *MSR1* and other M2 markers, instead we observed a notably higher expression of HLA-II genes, chemokines and M1 markers such as *FCGR3A* indicating a rather pro-inflammatory phenotype (Supplementary Fig. 12E).

### Analysis of biomarkers for immunotherapy

Among 22 evaluable patients treated with immune checkpoint inhibitors, two patients (1 adenocarcinoma NOS with *EWSR1*-fusion, thus likely a clear cell carcinoma and 1 carcinoma-ex-pleomorphic adenoma) achieved clinical benefit (stable disease for more than 6 months). Evaluable results from CIBERSORT deconvolution analysis ($n = 10$, including 1 with clinical benefit, P-95) revealed the highest T-cell to Macrophage ratio in the patient with clinical benefit (Fig. 6A). Additional RNA-sequencing was performed on 6 SGC patients treated with immune checkpoint inhibitors (clinical benefit in one myoepithelial carcinoma) (Fig. 6A, Supplementary Fig. 13A). Evaluable CIBERSORT results ($n = 4$) were integrated with prior data and further confirmed previous findings (Fig. 6A). Independent validation in a second cohort of evaluable patients with pre-treatment sequencing ($n = 14$) also revealed clinical benefit in the patient with the highest T-cell infiltration (Fig. 6B, Trial_ID_44). These results were additionally confirmed by an analysis of post-treatment sequencing in the same cohort ($n = 8$, Supplementary Fig. 13B).

We additionally performed an IHC analysis of T-cell/Macrophage ratio in a third, independent cohort of 5 patients treated with immune checkpoint inhibitors (clinical benefit in a patient with adenocarcinoma NOS; Supplementary Fig. 13C–F) and again identified the highest T-cell to macrophage ratio in the patient with clinical benefit (Supplmentary Data 2, Fig. 6C; exemplary images of stainings in the patient with clinical benefit are depicted in Fig. 6D, E). Evaluable patients with clinical benefit had a moderate immune cell infiltration and medium/high TMB in the context of the whole cohort (Fig. 6A, B and Supplementary Fig. 13A, B). We could not observe significant differences in the T-cell to Macrophage ratio or in T-cell proportions between samples that received ICI before sequencing or after (Fig. 6F, G).

To identify additional treatment targets, a set of 43 immune checkpoint molecules was analyzed for differential expression (Fig. 7A). Most of the immune checkpoints were enriched in the immune-high cluster. We could observe an overexpression of inhibitory members of the immunoglobulin superfamily *VTCN1* (syn: B7-H4, L2FC = 2.2, p.adj = 2e-08***) and CD160 (L2FC = 3.5, p.adj = 2e-19***) in ACC compared to other entities (Fig. 7B). Several immune checkpoints were significantly under-expressed in ACC (p.adj < 0.001*** and L2FC < −1.5), such as inhibitory members of the immunoglobulin superfamily *PD-1*, its ligand *CD274*, and *TIGIT*, the co-stimulatory molecules *TNFSF8* and *TNFRSF9 (4-1BB)*, and *NT5E (CD73)* (Fig. 7A).

We additionally analyzed mRNA expression of a comprehensive list of potential targets of T-cell-receptor based immunotherapy strategies. Expression of any potential treatment target was identified in the majority of samples. Expression clusters of target groups were

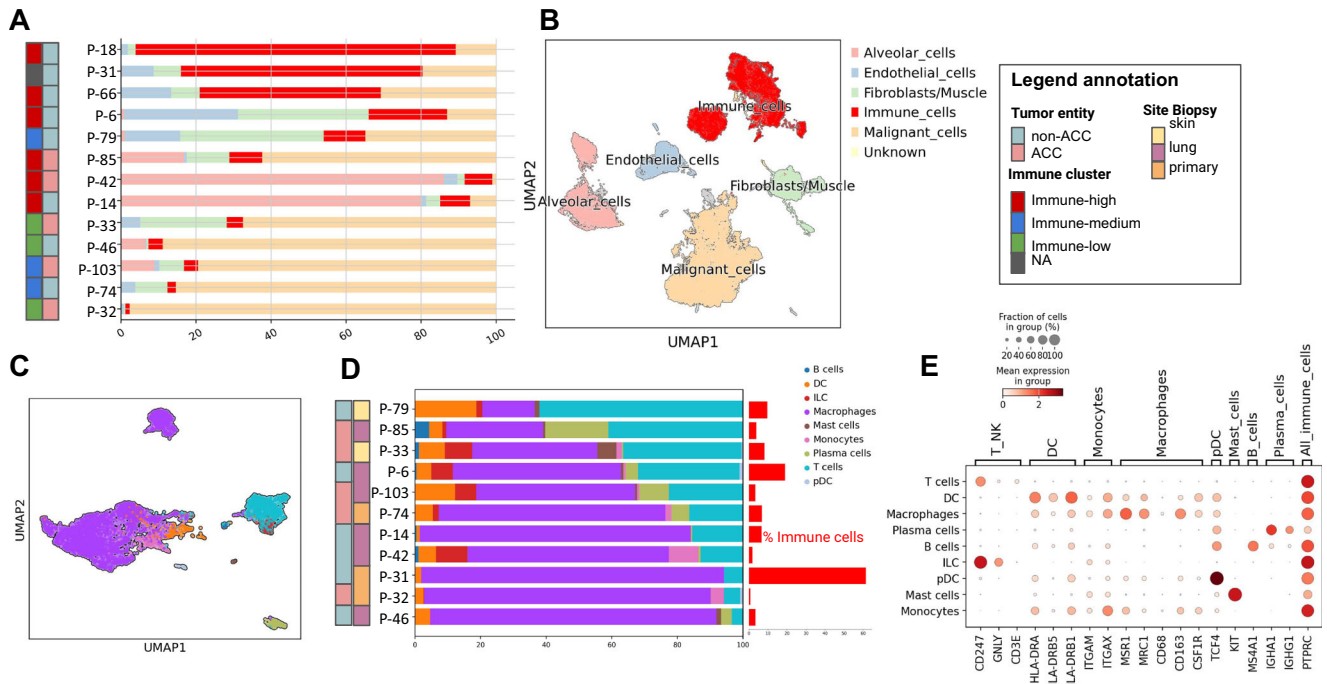

**Fig. 4 | TIM composition analysis based on single nuclei data. A** Proportions of major cell-types in single nuclei data revealed a higher number of immune cells (red) in tumors previously classified as immune-high (legend is provided in panel B). Samples are annotated by tumor entity and bulk-based immune cluster (see legend annotation). P-31 did not have evaluable bulk data and therefore lacked immune cluster annotation. **B** UMAP plot of integrated data, annotated by major cell types. **C** UMAP of immune cells (*n* = 11, after exclusion of lymph node metastasis samples) annotated by major immune cell populations. **D** Proportions of immune cell populations in single nuclei data revealed a predominance of macrophages (legend is provided in panel D). Samples were annotated by tumor entity and biopsy site (see legend annotation). The bars on the right represent the proportion of immune cells in the sample. **E** Expression of selected immune cell markers in annotated immune cells.

identified, e.g. *CTAG1B* (*NY-ESO-1*) and *MMP7* were found to be significantly overexpressed in ACC (L2FC = 4.7, p.adj = 1.9e−10*** and *L2FC = 3, p.adj* = 1.2e−11***). Around 25% of the cohort had high expression of some or multiple genes of the MAGE family (Supplementary Fig. 14A).

Protein expression of *VTCN1* was validated by IHC and correlated with bulk RNA expression in 15 samples (Fig. 7C, D and Supplementary Fig. 14B, C). IHC intensity multiplied by percentage of positive cells was correlated with *VTCN1* expression (*Spearman, rho = 0.82, p* = 1.9e−4***, Fig. 7D). *VCTN1* expression was further confirmed in tumor cells of ACC samples in single-nuclei RNA seq data (Fig. 7E). No association between *VTCN1* expression and different immune clusters in ACC was seen (Fig. 7A). Further analysis in single nuclei data demonstrates that *VTCN1* was overexpressed in luminal-like cells and showed decreased expression in myoepithelial-like cells within ACC samples (Fig. 7E–G). A predominance of luminal-like cells was identified in ACC1 (*n* = 1), whereas ACC2 (*n* = 3) had more myoepithelial-like cells. These differences in cell composition might mediate differences in *VTCN1* expression between ACC1 and ACC2 samples[15], however, the expression in both ACC clusters was remarkably higher than in non-ACC (Fig. 7B). These findings indicate that *VTCN1* is a suitable marker for ACC specifically.

## Discussion

Effective immune therapies are lacking for patients with advanced SGC. Response rates with PD-1 or PD-1/CTLA-4 directed therapies are low and range from 4-16% in prospective clinical trials[7,9]. Consequently, the use of immune checkpoint inhibitors has not been recommended outside of clinical trials in current guidelines[12]. In order to identify potential biomarkers and therapeutic strategies for immunotherapy in these hard-to-treat tumors, we analyzed a large cohort of advanced SGC using bulk and single-cell sequencing data.

An inflamed TIM could be observed in only a subset of samples. Immune desertion was most pronounced in Intercalated-duct derived (ID) histologies, including myoepithelial carcinoma and ACC, compared to excretory-duct derived (ED) histologies, in line with previous analyses in SGC[13,23]. However, prior data are based on early tumor stages and, as such, are at risk of bias towards more favorable disease subgroups. The provided analysis of advanced tumors yields multi-layered data in a potential intention-to-treat cohort that might differ biologically from early-stage tumors. The observed differences between primary tumors and lung metastases in ACC support this notion. Pulmonary metastases have been associated with a more favorable clinical course in ACC[24] and are therefore likely linked to different ACC subgroups.

These results highlight the heterogeneity between tumor subtypes both within ACC and SGC overall, in line with data from a reanalysis of early-stage SGC gene expression data, where an association between cell-of-origin and tumor inflammation was shown[23]. Several mechanisms might contribute to these phenotypic differences. Inflammation was associated with TMB mostly in non-ACC histologies in our analyses. This is likely caused by a uniformly lower TMB in ACC, limiting a statistical association. However, the molecular profile of inflammation-high ACC did not differ significantly from the majority of ACC, therefore, the mechanisms behind tumor inflammation in a small ACC subset remain not entirely understood. Within ACC, different subgroups of ACC1 and ACC2 have recently been proposed[4]. We were able to validate the prognostic relevance and also link these findings to differences in tumor inflammation. Our results further suggest that these differences are mediated by differences in the antigen processing machinery. In contrast, non-ACC samples, especially an ED-like cell-of-origin, often harbor complex genomic alterations and higher tumor inflammation (e.g., APOBEC-mutational signature in mucoepidermoid carcinoma samples or TMB-high in salivary duct carcinoma).

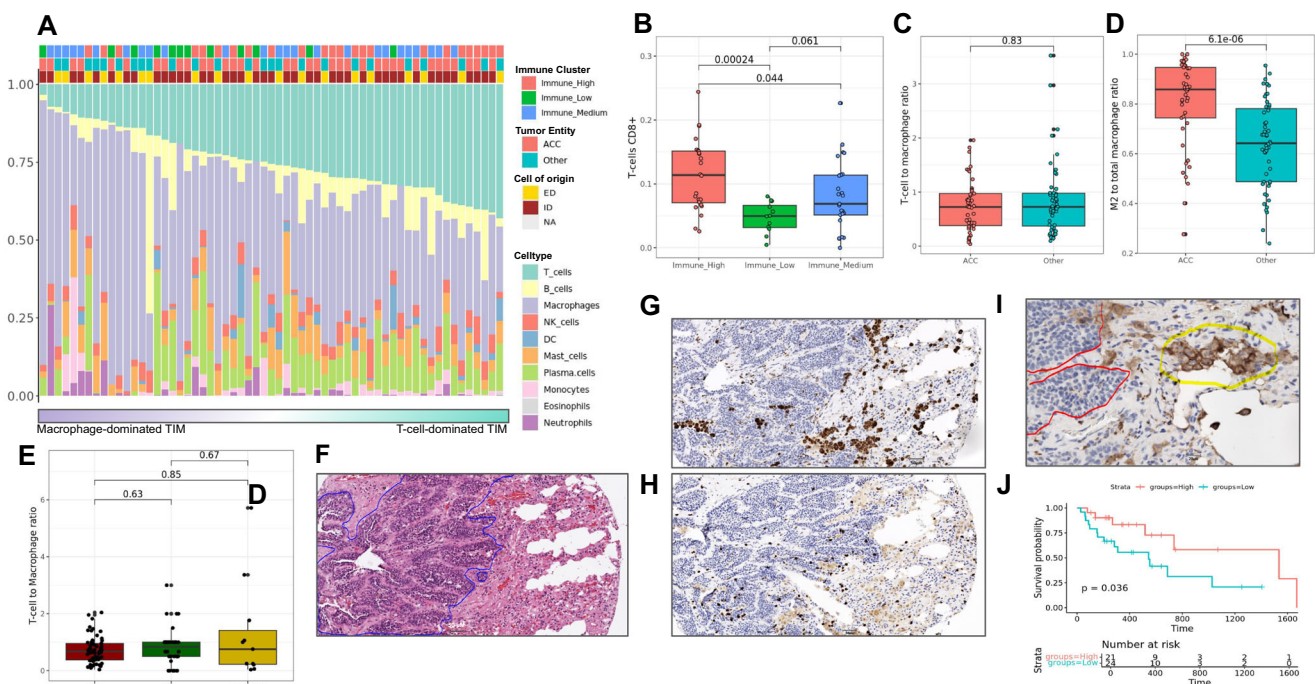

**Fig. 5 | Macrophages dominate TIM in advanced SGC. A** Deconvolution results for 61 samples with evaluable results. Barplot shows the proportions of major immune cell subpopulations. Samples were ordered by T-cell to macrophage ratio and annotated by tumor entity, cell of origin (ID/ED group), and immune cluster. **B** Deconvolution results revealed significantly different proportions of CD8 T-cells between previously identified immune subgroups (p-value filtered, Immune-high $n = 24$ median = 0.11, iqr = 0.08, max = 0.24, min = 0.03; Immune-medium $n = 25$ median = 0.07, iqr = 0.06, max = 0.23, min = 0.00; Immune-low $n = 12$ median = 0.05, iqr = 0.03, max = 0.08, min = 0.00) (Wilcoxon test, two-sided). **C, D** In an integrated analysis including previously published studies, the overall T-cell/Macrophage ratio did not differ between ACC ($n = 47$, median = 0.72, iqr = 0.59, max = 1.96, min = 0.04) and non-ACC ($n = 51$, median = 0.73, iqr = 0.61, max = 3.52, min = 0.10) samples (**C**), whereas a higher M2-macrophage/overall macrophage ratio was observed in ACC ($n = 47$, median = 0.86, iqr = 0.20, max = 1.00, min = 0.28) compared to non-ACC ($n = 51$, median = 0.64, iqr = 0.29, max = 0.95, min = 0.24) samples (**D**) (integrated, p-value and TNM filtered data, $n = 98$) (Wilcoxon test, two-

sided). **E** A low T-cell/macrophage ratio was confirmed across different analytic modalities including bulk sequencing ($n = 61$, median = 0.68, iqr = 0.58, max = 2.04, min = 0.04), single-cell sequencing ($n = 11$, median = 0.75, iqr = 1.19, max = 5.72, min = 0.04), and immunohistochemistry ($n = 40$, median = 0.83, iqr = 0.50, max = 3.00, min = 0.00) (Wilcoxon test, two-sided). **F** Representative H&E staining of a tumor sample (ACC). Highlighted in blue is the invasive tumor front. **G** Same sample as in panel F showing the presence of macrophages (CD68 staining) (Same scale as figure F). **H** Same sample as in panel F and G showing the presence of CD8 T cells (Same scale as figure F and G). **I** Representative image of M2-macrophages clustering towards the tumor edge in the tumor immune microenvironment in a SGC samples. M2-like macrophages (CD163) are highlighted by yellow, whereas the red line shows the tumor front. **J** Survival plot of the samples with the highest and lowest T-cell proportion (upper- and lower quartile of deconvoluted T-cell proportions) shows a nonsignificant trend towards improved survival with higher T-cell proportions (log-rank test, two-sided).

These differences could cause the higher efficacy of immune checkpoint inhibitors in non-ACC compared to ACC subtypes, observed in this cohort as well as clinical trial data[9]. Thus, individual SGC patients, especially with an ED-like histology, might benefit, and an individual molecular tumor profile should be evaluated before a treatment decision is made[12]. These individual considerations are likely more important than simple assignment to one of the three identified inflammation groups. However, these groups are able to distinguish between existing T-cell inflammation, the presence of immune effector cells, and immune desertion. Expression of immune checkpoints across these groups also suggests to differentiate between immune-high and immune-deserted tumors. Still, overall response rates to immune checkpoint inhibitors are also low for non-ACC samples. We assessed the TIM to identify potential additional mediators of resistance to current immunotherapy strategies.

Identifying immune cell compositions from bulk sequencing is challenging. Using single-cell transcriptome-validated bulk deconvolution analyses, we identified a myeloid-dominant tumor immune microenvironment (TIM) in both ACC and non-ACC SGCs. Macrophages, particularly M2-polarized macrophages, are key drivers of immune suppression and resistance to immune checkpoint inhibition[25]. The high M2 macrophage content in advanced SGCs may thus contribute to ICI resistance. Comparative analyses with other

TCGA datasets further suggest that relative macrophage predominance is particularly relevant in SGC.

A high T-cell content was associated with clinical benefit from immune checkpoint inhibitors (ICI) in individual patients in two independent cohorts. However, in both cohorts, only a subset of samples could be confidently analyzed via deconvolution, which limits the statistical power of findings. Response to ICI is multifactorial and influenced by several factors, such as TMB or PD-L1 expression[26]. Thus, these results need to be viewed with caution before translation into clinical practice. Yet, they justify further validation of ICI treatment in SGC subgroups and investigation of the T-cell to macrophage ratio as an additional potential predictive biomarker. Macrophage-directed agents should be further investigated either alone or in combination with other immunotherapies for patients with SGC. Mechanisms underlying the high macrophage to T-cell ratio in SGC are currently unclear. We were able to assess the immune microenvironment of healthy salivary glands, which did not reveal an increased macrophage content. Most macrophages in SGC exhibited a rather M2-like phenotype, however, our single-cell analysis of the macrophage population in healthy SG revealed a predominantly inflammatory phenotype. Nevertheless, some M2-like macrophages were still present within the tissue. This coexistence suggests that macrophages exist along a dynamic spectrum rather

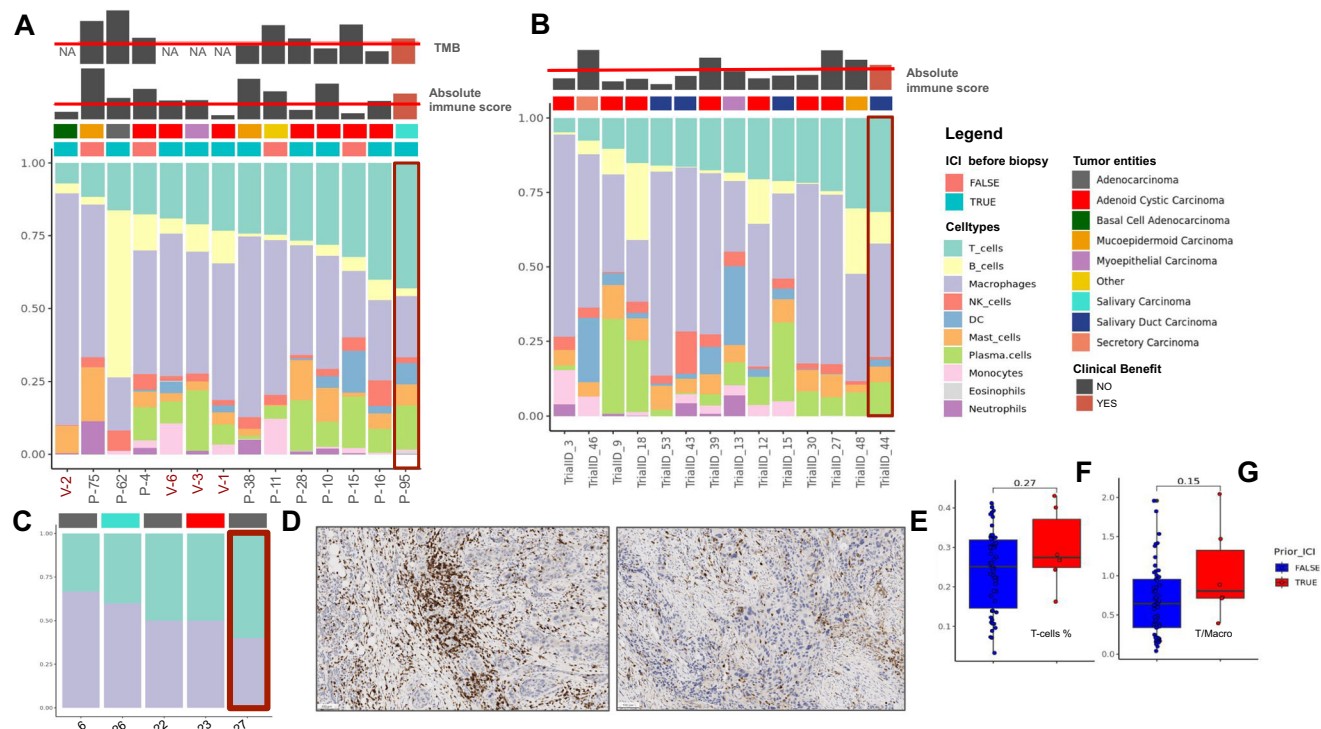

**Fig. 6 | Clinical benefit in individual patients with high T-cell infiltration.** **A** Deconvolution results for 14 samples of patients treated with immune checkpoint inhibitors from the MASTER cohort, also including additionally integrated data from a validation cohort of 4 patients (red font). Barplot shows proportions of major immune cell subpopulations. Samples were ordered by T-cell to macrophage ratio and annotated by tumor entity and therapy status (ICI prior to sequencing). Top bars indicate the absolute immune score and TMB. The red lines depict median scores. Patients achieving a clinical benefit were marked in red. **B** Deconvolution results for 14 samples from the Vos et al pre-treatment cohort. For details see panel A. TMB results were missing for this cohort. **C** Additional validation in a separate cohort of 5 patients treated with ICI using immunohistochemistry. Staining intensities of CD68 and CD3 were normalized to sum up to 1. Samples were annotated by tumor entity and sorted by T-cell proportions. Clinical benefit was observed in the patient with the highest T-cell/macrophage ratio (highlighted in red). **D** Immunohistochemical CD3 staining revealed the presence of T-cells in the patient achieving a clinical benefit from **C**. **E** Immunohistochemical CD68 staining depicting some presence of macrophages in the same patient. Deconvoluted, *p*-value filtered T-cell proportions (**F**) and T-cell to macrophage ratios (**G**) in samples that received ICI prior to sequencing (*n* = 6, median T-cell proportion = 0.28, iqr = 0.12, max = 0.43, min = 0.16; T-cell/macrophage ratio = 0.81, iqr = 0.61, max = 2.04, min = 0.39) or did not receive ICI before sequencing (*n* = 54, median T-cell proportion = 0.25, iqr = 0.17, max = 0.41, min = 0.03; T-cell/macrophage ratio = 0.65, iqr = 0.62, max = 1.96, min = 0.04) revealed no significant changes in tumor immune microenvironment composition after ICI treatment, although a modest increase in T-cell levels was observed (*Wilcoxon test, two-sided*).

than fitting into strictly defined M1 or M2 categories. The presence of distinct immune checkpoint molecules on SGC suggests close interaction with the TIM and might represent potential therapeutic targets. High expression of *VTCN1* (B7H4) was identified in ACC, thus validating previous findings in cohorts with mostly earlier-stage ACC[15]. In contrast to these previous results, we only found a minor difference in *VTCN1* expression between ACC1 and −2 subtypes, which could therefore be less pronounced in more advanced disease stages. However, significantly different *VTCN1* expression was identified between luminal- and myoepithelial cells within ACC, which might impact the therapeutic efficacy of *VTCN1*-directed agents and explain previously reported differences between ACC subtypes. *VTCN1* has been shown to negatively regulate T-cell immune response[27] and was negatively associated with tumor-infiltrating lymphocytes and *PD-L1* expression in breast cancer[28]. Hence, over-expression of *VTCN1* on ACC cells could contribute to the low immunogenicity in ACC and poor response to ICI and might represent a potential treatment target for advanced ACC. In addition to immune checkpoints, expression of target antigens for immunotherapy strategies was identified in a relevant subset of samples, including a histology-predominant expression of *NY-ESO1* in ACC, in line with previous reports[29]. Advanced SGC should therefore be incorporated in ongoing trials, such as T-cell-receptor-based immunotherapies.

In conclusion, an inflamed TIM can be observed in a subset of advanced SGC with ACC and other ID-like histologies showing significantly less inflammation. TMB and specific mutational signatures in non-ACC and antigen processing in ACC are potential contributors to these observed phenotypes. ICI and T-cell-receptor-based therapies should be further investigated in biomarker-stratified SGC. Among these targets, significant *VTCN1* over-expression in advanced ACC could represent a possible treatment option. Furthermore, TIM cell compositions are characterized by macrophage predominance, representing an additional potential treatment target, whereas a high T-cell/macrophage ratio should be further investigated as a predictive biomarker for ICI. These results further support the development of biomarker-based immunotherapy strategies in advanced SGC.

While our study addresses key gaps in the literature, certain limitations must be acknowledged. The heterogeneous nature of the cohort—encompassing diverse subtypes, biopsy sites, and treatment histories—introduces variability that complicates the attribution of specific findings to individual subtypes or populations. This is especially important for rare SGC subtypes, where sample size does not allow for a generalization of findings. Furthermore, multiple covariates, such as prior treatments, may influence the TIM, potentially masking or mimicking significant associations. Despite these limitations, our study provides important insights, including the provision of data on rare SGC subtypes and a focus on recurrent and metastatic

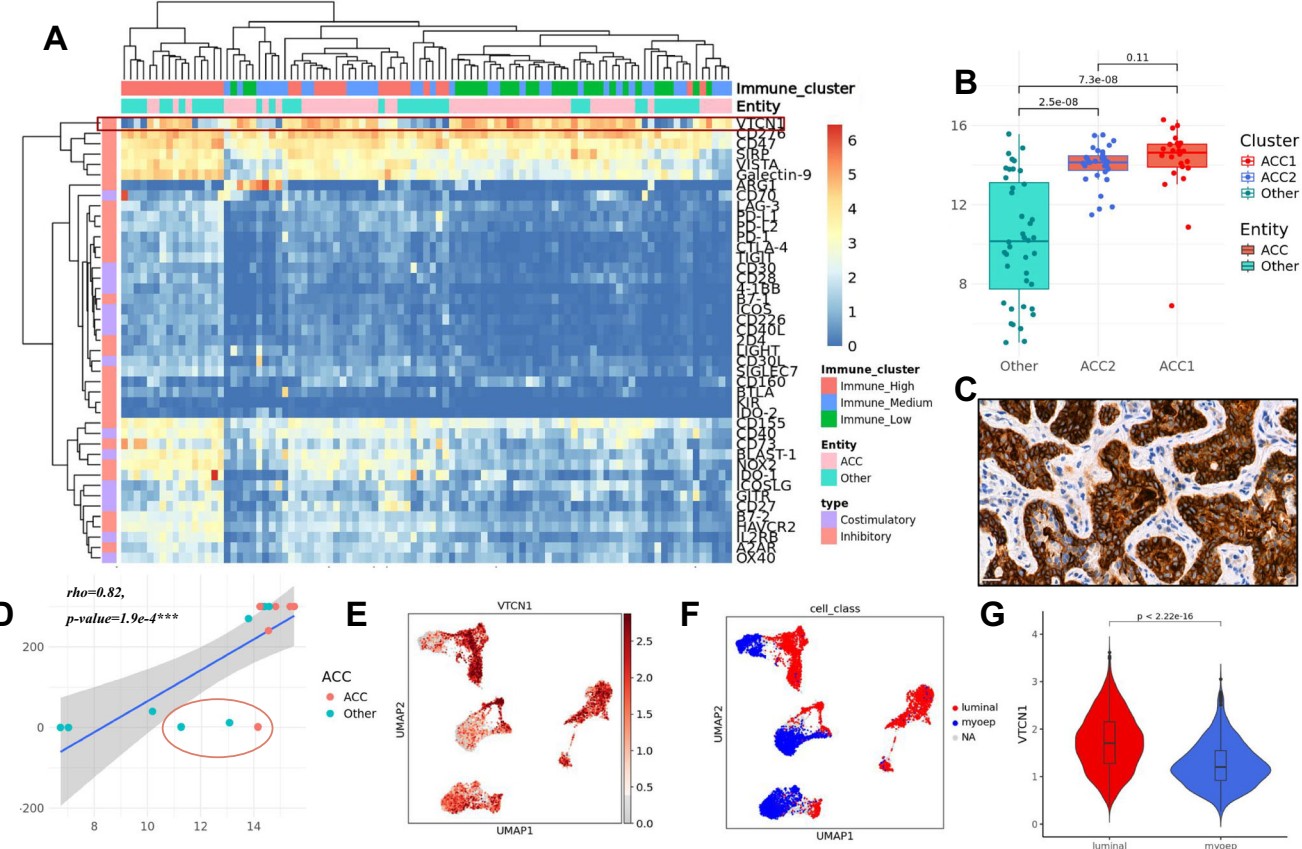

**Fig. 7 | Analysis of biomarkers for immunotherapy. A** Expression matrix based on log TPM values of selected immune checkpoints. The checkpoints were annotated on the left as co-inhibitory or co-stimulatory. The samples were annotated based on the tumor entity and the immune-cluster assignment. *VTCN1* was highlighted in red. **B** Boxplot of *VTCN1* expression (here vst values) in ACC1 ($n = 25$, median = 14.60, iqr = 1.16, max = 16.3, min = 6.19), ACC2 ($n = 32$, median = 14.10, iqr = 0.75, max = 15.5, min = 11.5), and non-ACC (n = 38, median = 9.92, iqr = 6.06, max = 14.9, min = 0.71) showing significantly increased *VTCN1* expression in ACC compared to non-ACC samples and a nonsignificant trend towards increased *VTCN1* expression in ACC1 (*Wilcoxon test, two-sided*). **C:** Exemplary staining of *VTCN1* in an ACC sample reveals strong staining intensity on tumor cells. **D:** *VTCN1* expression by RNAseq

was significantly correlated with *VTCN1* staining intensity by IHC (intensity * percentage positive cells), with the exception of 3 outliers (highlighted in red). The shaded area represents the 95% confidence interval of the linear model. **E:** Expression of *VTCN1* in ACC malignant cells. UMAP shows non-integrated data. Cells are clustered by donor. **F:** UMAP of malignant cells in ACC colored by assigned cell type (myoepithelial- or luminal-like) revealed an association between *VTCN1* expression and cell type. **G:** Violin plot with expression of *VTCN1* in myoepithelial- vs. luminal-like malignant cells showed significantly higher expression in luminal ACC cells (luminal: n = 2615, median = 1.70, iqr = 0.88, max = 3.61, min = 0.38; myoepithelial: $n = 1033$, median = 1.20, iqr = 0.63, max = 3.05, min = 0.26). Cells expressing no *VTCN1* were removed (*Wilcoxon test, two-sided*).

tumors representing an intention-to-treat population for systemic therapies. The use of multi-omics approaches—combining WES, WGS, RNA-Seq, and scRNA-seq—alongside clinical data, enables a comprehensive analysis of the TIM. By contextualizing advanced SGC TIM features with comparisons to healthy salivary tissue, other cancer types, and previously published cohorts, we offer unique insights into the immune dynamics and biology of advanced SGC.

## Method
### Patient identification
The research complies with all ethical regulations (Ethics Committee Medical Faculty Heidelberg, Germany, S-206/2011, Ethics Committee Charité Berlin, Germany, EA1/305/21; see below).

Patients with ACC and non-ACC SGC from the multicentric national DKFZ/NCT/DKTK MASTER (Molecularly Aided Stratification for Tumor Eradication Research, Clinicaltrials ID NCT05852522) program were included in the analysis. The MASTER program applies comprehensive molecular diagnostics to inform the care of adult patients with incurable cancers. MASTER inclusion criteria were advanced solid tumors of a rare histology or younger age (<51 y), no available standard therapy, good general condition (ECOG < 2), as well as available fresh-frozen tumor tissue. Participants are not

compensated. The MASTER program was approved by ethics committees at all participating sites (Lead Heidelberg, S-206/2011). In a separate retrospective analysis, patients with salivary gland cancer treated at Charité - Universitätsmedizin Berlin were identified through an analysis of the clinical and pathological documentation system.

Written informed consent was obtained from all participants in the MASTER program. Ethics approval for retrospective analysis of tumor samples from patients not participating in the MASTER program was obtained separately (Berlin, EA1/305/21). Sex was considered in the study design and self-reported by participants. Clinical data were extracted from clinical documentation of the primary care facility detailing the patient history, as available.

### Immunohistochemistry
Formalin-fixed and paraffin-embedded (FFPE) surgical specimens of the patients were used to prepare slides for immunohistochemistry (IHC). IHC was performed on tissue microarrays (TMAs) with two cores for each case, according to standard procedures. As counterstaining, Hematoxylin was used.

For the detection of the immune cells we used a polyclonal antibody against *CD3* for T-cells (solution 1:100, Dako) and monoclonal

antibodies against *CD8* (clone *C8/144B*, solution 1:100, Dako) for cytotoxic T-cells, against CD4 (clone 4B12, solution 1:20, Leica) for T-helper cells, against *CD20* for B-cells (clone L26, solution 1:750, Dako), against CD68 for macrophages (clone PG-M1, solution 1:200, Dako), against CD163 (clone NCL-L-Cd163, solution 1:400, Leica) for M2-macrophages, against *FOXP3* for T-regs (clone 236A/E7, solution 1:200, Abcam plc., Cambridge, UK) and *PD-L1* (clone E1L3N, solution 1:200, Cell signaling). All analyses were performed on the Leica Bond Master.

For the detection of *VTCN1*, we used the recombinant Anti-B7H4 antibody (clone EPR23665-20, solution 1:100, Abcam plc., Cambridge, UK). A positive expression for *CD3, CD4, CD20, CD68, CD163, and CD8* was defined by a medium to strong intensity of membranous staining. A positive expression for *VTCN1* was defined by a medium to strong intensity of membranous staining of the tumor cells. IHC intensities were classified as 0 for no staining, 1 for weak intensity, 2 for medium intensity, and 3 for strong intensity by a board-certified pathologist. Immunohistochemistry staining was performed once on each sample for each selected marker.

### Exome, genome, and bulk RNA seq sequencing
Bulk sequencing on fresh-frozen tumor tissue within the MASTER program was performed as previously described[30]: DNA and RNA from tumor and DNA from matched blood samples were isolated using AllPrep Mini or Universal Kits (Qiagen). After library preparation (SureSelect Human All Exon, Agilent; TruSeq RNA Sample Preparation kit V2, Illumina), whole-exome and RNA-paired-end sequencing (2 × 151 bp; 2 × 101 bp) was performed with various HiSeq instruments (e.g. HiSeq 4000 and NovaSeq 6000; Illumina).

### Exome, genome and bulk RNAseq data processing
*Fastq* files were trimmed using *bbduk* by removing adapter sequences and then mapped using *bwa-mem* algorithm (version 0.7.17) onto the GRCh38 genome (GRCh38.d1.vd1, primary assembly with decoys and viral sequences). In case of RNA-seq data, alignment and gene expression quantification were performed with salmon (version 1.4.0, transcriptome version: GENCODE 33). Counts were normalized to gene length (TPM) and log-transformed for visualization. To compare gene expression between samples (for example, in a box-plot) and to run GSVA, counts were "variance stabilized" via vst transformation using the DeSeq2 package in R[31]. Aligned exome and genome data was passed through a somatic variant calling pipeline. SNPs and short indels were detected by the *Mutect2* algorithm[32] using in-house panels of normals. The resulting vcf files were annotated via *jannovar* (version 0.26) and *vep* (version 102). Gene fusion products were detected with *arriba*[33] (version 2.3.0). CNVs were computed with *Ascat*[34], using both exome and genome data. A CNV was labeled as amplification if the copy number was larger than 4 and as deletion if the copy number was 0. The chromosomal abberation index (CAI) was defined as the length of the altered genome (affected by CNVs) divided by the total length of the genome. Other datasets used in this study were retrieved from the TCGA database and the NCBI sequence read archive (SRA). For TCGA datasets *STAR* counts were used and transformed. Linxweiler et al[13] and Vos et al[9] comprised *fastq* files from SRA which were processed in the same manner as our data. Bulk RNA-seq raw data from healthy SG (salivary gland) tissue were retrieved from 3 different publications[20-22]. Only samples from healthy and adult tissue were selected and processed in the same manner as our data. The final count matrix was then batch corrected using the study as a batch.

### Bulk RNAseq downstream analysis
Differential expression was performed using the R package *DeSeq2*[31]. DEGs and PCA loadings were tested for functional enrichment against gene sets from MSigDB (release 2023 v1) using the *tmod*[35] package.

The AUC for each enrichment was calculated by ranking genes from the gene set to be tested based on their relevance to a specific module and then assessing how well these ranks separate module-associated genes from non-associated ones

Sample-wise gene set enrichment was performed with GSVA implemented in the R package *gsva*[36]. A set of several functional scores for immune infiltration published in four independent publications was used to cluster the patients. Gene signatures were selected based on similar sizes, frequent citations, and association with tumor immunity. This set comprises a 6-gene T-cell signature from Danaher et al., a 6-gene IFNG signature from Ayers et al., an 18-gene immune signature from Ayers et al., a 7-gene APM (Antigen processing machinery) signature from Senbabaoglu et al. a 17-gene cytotoxicity signature from Bindea et al. and a 20 ICR (Immunologic Constant of Rejection) gene signature from Roelands et al.[37-41]. The overlap between these signatures was low (max. Jaccard's coefficient 0.2). Furthermore, scores for several immune infiltrates were computed based on the set of immune infiltrate signatures from Bindea et al.[40]. All signatures are listed in Supplementary Data 3.

As an additional validation, other scores were computed: the cytotolic score[42], the ESTIMATE Immune score[43], the CIBERSORT absolute score[44], and the x-cell immune score[45]. CIBERSORT was used both in absolute and relative modes to get the absolute and relative proportions of immune cells, respectively. For the validation of the CIBERSORT results in the immune checkpoint inhibitor (ICI)-cohort,t we analyzed 6 additional samples from the MASTER cohort and the published RNA-seq cohort from Vos et al.[9]. These analyses were performed using the *immunedeconv* package in R[46].

GSVA was applied using a Gaussian kernel on variance-stabilizing transformed counts (*vst*). The input for the *immunedeconv* methods, such as CIBERSORT and x-cell were TPMs. Results from different deconvolution methods were compared to single-cell results and IHC via simple correlation, the best method was chosen for further analyses. Matrices with GSVA scores of immune-cells/infiltrates were clustered via hierarchical clustering with ward linkage and $k = 3$. Immune infiltration scores were correlated against each other, and the $p$-values were corrected for multiple testing (bonferroni-holm method).

In the case of the integrated analysis with other cohorts, samples from tumor stage <3 were filtered out (this filtering step was performed in order to analyze only advanced tumor samples) and the filtered combined expression matrix was batch corrected with the *limma* package in R before computing the GSVA scores.

### Integration and statistical testing of clinical and molecular data
We analyzed the relationships between several clinical parameters and response variables such as IFNG-score or t-cell to macrophage ratio using one- or two-way ANOVAs. For non-normally distributed response variables, log transformations were applied to meet model assumptions where appropriate. In cases where transformations were insufficient, alternative methods such as non-parametric ANOVA (Kruskal-Wallis) were employed to ensure robust statistical inference. To test differences between groups in any other settings, a non-parametric, unpaired test such as the Wilcoxon test was used. P-values were corrected using the Holm-Bonferroni method.

Survival analyses were performed with R package *survminer* using only overall survival. Samples were divided into "high" or "low" for survival analysis by using the upper quartile and lower quartile of the respective immune score.

Two molecular ACC subgroups were previously described by Ferrarotto et al.[4]: ACC1 and ACC2. ACC1 subtype is characterized by solid histology and poorer prognosis, whereas ACC2 subtype has a better prognosis and predominantly cribriform tubular forms. ACC1-ACC2 score was calculated based on *MYB-TP63* expression as described by Ferrarotto et al[4]. If the score was > 0, the sample was labeled as ACC1, otherwise as ACC2.

Different entities were labeled as ID (Intercalated-duct)- or ED (Excretory-duct)-derived based on a cut-off on median SOX10 expression and prior literature, when applicable[23,47].

## WGS and WES downstream analysis

Annotated variants retrieved from exome and genome data were used for the calculation of the tumor mutational burden (TMB), mutational landscape analyses, and detection of mutational signatures. TMB was calculated as the number of non synonymous short variants per MB sequence. Variants used for TMB had at least 5% VAF, 10x coverage for the normal allele and 20x or 50x coverage for the mutant allele (depending on the data source: WGS or WES). VCF files were converted to MAF files by the *vcf2maf* tool (version 1.6.21) and analyzed by the *maftools* package[48] in R. For mutational landscape analyses, only mutations in the coding sequence were used, whereas for mutational signatures, all variants were used.

The oncoprint matrix was produced using all alterations (SNVs, CNVs, and fusions) in a set of selected genes, known to be affected in SGC[49]. Mutational signatures were computed with MuSiCal[50]. Exposures were calculated against the newest version of SBS mutational signature catalog (v3.4)[51]. To test for positive selection the ratio of missense and synonymous mutations within the coding sequence of immunotherapy-relevant genes was computed gene-wise. If the ratio > 1 there is a positive selection for the respective mutation(s). Tumor purity and ploidy were assessed with *Ascat*[34] based on WGS and WES.

## Tissue dissociation, nuclei preparation, and single nuclei sequencing

Each sample of fresh frozen tumor tissue was suspended in NP-40 lysis buffer (10 mM Tris-HCl (pH 7.4); 10 mM NaCl; 3 mM $MgCl_2$; 0.01% NP-40; 1 mM DTT; 2% BSA; 1U/µl RNAse inhibitor; Complete EDTA-free protease inhibitor) in a 1.5 ml Eppendorf tube and disrupted with a plastic pestle. The suspension was incubated on ice for 5 minutes, filtered through a 70 µm pre-separation strainer, and centrifuged at 4 °C. The supernatant was removed, the nuclei were resuspended in nuclei wash buffer (PBS; 1% BSA; 0.4U/µl RNAse inhibitor), and 3.5 µl DAPI was added, followed by incubation on ice. The mixture was filtered through a 40 µm Flowmi cell strainer and sorted with a 100 µm nozzle in an Eppendorf tube containing 200 µL sort buffer (PBS; 2% BSA; 2U/µl RNAse inhibitor). Single nuclei libraries were generated according to the Chromium Next GEM Single Cell 3′ Reagent Kits v3.1 (Dual Index) user guide (CG0003154) by 10x Genomics. Gel bead-in-emulsions (GEMs) were created using the Chromium Controller. In this step, individual nuclei were encapsulated with a gel bead in a droplet, where barcoding occurs. The barcoded RNA was reverse transcribed into cDNA and amplified. The amplified cDNA underwent fragmentation, end-repair, A-tailing, adaptor ligation, and sample index PCR to create the final library. The prepared library was sequenced on a NovaSeq 6000 instrument (Illumina).

## Single nuclei data processing and downstream analysis

Generated *fastq* files were aligned with *cellranger* (version v7.1.0), which outputs count matrices for each sample. *Cellbender*[52] was then used to filter count matrices and remove background noise. Expected cells and total droplets were estimated from the cell ranger quality control for each sample. The count matrices were further processed using the *scanpy* package in Python. In the case of published data for healthy SG[22], the filtered count matrix in.h5 format was used and processed further in the same manner as our data.

After cellbender filtering, additional filtering steps were performed. Only cells with at least 1000 UMIs and at least 300 expressed genes were kept. Cells with more than 20% mitochondrial and 10% ribosomal counts were removed. *scDBLFInder* was used for doublet detection[53], which was around 10% in the whole dataset.

Counts were normalized via size factor normalization[54]. For PCA, only 3000 to 5000 highly variable genes (depending on the analysis)

were used. In certain analyses, counts from mitochondrial and ribosomal reads, as well as from *MALAT1*, were removed in order to facilitate the interpretation of results. Furthermore, a total of 50 PCs were used for integration with *Harmony*[55]. After assessment of 3 different integration methods, we found *Harmony* to be the best suited for our data. Harmony embeddings were used for knn-graph construction, UMAP, and clustering. For clustering, we used the *leiden* algorithm[56] with 0.5 resolution. Cell clusters were annotated manually based on markers retrieved from the literature[57] and the protein atlas[58] (proteinatlas.org). Immune cells were annotated automatically using published *celltypist*[59]. Malignant cells were identified based on copy number variants (*inferCNV* of the Trinity CTAT Project. https://github.com/broadinstitute/inferCNV) and based on the occurrence of somatic SNVs known from WES/WGS in the single-cell RNAseq reads with CCISM[60]. Malignant cells in ACC were further classified into myoepithelial- and luminal-like following a similar approach to the one used by Parikh et al.[61]. Luminal and myoepithelial scores were calculated and subtracted. If the difference was below or above a 0.3 threshold, the cells were labeled either as luminal or as myoepithelial.

## Statistics & Reproducibility

**Study design and sample size**. No statistical method was used to predetermine sample size. The study included all eligible patients with available sequencing data and/or immunohistochemical analyses from the DKFZ/NCT/DKTK MASTER cohort and the retrospective Charité - Universitätsmedizin Berlin cohort. Sample sizes were based on the availability of tumor samples and matched clinical data.

**Data exclusions**. Samples with insufficient quality, defined by low sequencing quality, incomplete clinical documentation, or technical failure in immunohistochemical staining, were excluded. Additionally, for integrated analyses involving multiple cohorts, samples from tumor stages below stage III from published cohorts were excluded to focus specifically on advanced disease.

**Randomization and blinding**. The experiments were not randomized. The investigators were not blinded to allocation during experiments or outcome assessment. Clinical outcomes and molecular data were analyzed retrospectively.

**Statistical analyses**. Statistical analyses were conducted using R (versions 4.1.0–4.3.0) and Python (Scanpy for single-cell analysis). Details about the specific methods and tests can be found in the respective methods section.

**Reproducibility**. Immunohistochemical staining experiments were performed once per marker, per sample, due to the limited availability of tumor material. Sequencing experiments were conducted once per patient biopsy, and data reproducibility was assessed by validating results across independent cohorts and multiple analytic modalities (bulk RNA-seq, single-nuclei RNA-seq, immunohistochemistry). We additionally provide raw sequencing data under restricted access as well as processed data, such as count matrices for single-cell and bulk RNA-seq data.

# Ethics approval and consent to participate

The MASTER program was approved by ethics committees at all participating sites (Lead Ethics Committee Heidelberg, S-206/2011). Written informed consent was obtained from all participants. Ethics approval for retrospective analysis of tumor samples from patients not participating in the MASTER program was obtained separately (Ethics Committee Charité Berlin, EA1/305/21).

## Reporting summary

Further information on research design is available in the Nature Portfolio Reporting Summary linked to this article.

## Data availability

The bulk sequencing data generated in this study have been deposited in the The European Genome-phenome Archive (EGA) with the accession code EGAS50000000809. The data are available under controlled access due to the sensitive nature of genome sequencing data, and access can be obtained by contacting the appropriate Data Access Committee listed for each dataset in the study. Access will be granted to commercial and non-commercial parties according to patient consent forms and data transfer agreements. We have an institutional process in place to deal with requests for data transfer and aim for rapid response time. Any further questions can be directed to the corresponding author. The clinical data generated in this study are provided in the manuscript and supplementary information. The count matrix for bulk data and filtered count matrices for each single nuclei sample are provided on GEO under the accessions GSE294016 (bulk) and GSE294017 (single nuclei). We provided source data for each of the main figures. Source data are provided with this paper.

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

## Acknowledgements

The MASTER program is supported by the NCT Molecular Precision Oncology Program, DKFZ, and DKTK. This project was supported by the Deutsche Forschungsgemeinschaft (No. RTG2424 CompCancer, EZ) and a Berlin Institute of Health Booster Grant (EZ, DB, DTR). DTR was a participant in the Berlin Institute of Health - Charité Clinical Scientist Program funded by the Charité - Universitätsmedizin Berlin and the Berlin Institute of Health. This work was supported by the ACC Research Foundation (DTR) and the Innovationsfonds des Gemeinsamen Bundesausschusses (innovation fund of the German federal joint committee), grant number 01VSF22041 (DTR). The authors thank the NCT/DKFZ Sample Processing Laboratory, the DKFZ Next-Generation Sequencing Core Facility, and the DKFZ Omics IT and Data Management Core Facility for technical support as well as the Core Unit Genomics - Single-Cell Technologies, Max Delbrück Center, Berlin. The results published here are in part based upon data generated by the TCGA Research Network: https://www.cancer.gov/tcga.

## Author contributions

E.Z. conceptualized and performed analyses, interpreted the data, and drafted and revised the manuscript; BvdE performed analyses and data acquisition and drafted the manuscript; I. Pi performed analyses and interpreted data and drafted and revised the manuscript; AP interpreted the data and revised the manuscript; K.K. acquired and interpreted data and revised the manuscript; A.M. performed analyses, interpreted data and revised the manuscript; P.H. acquired and interpreted data and revised the manuscript; C.H., F.K., I. Pr, M.B., C.H.B., S.K., M.V.T., D.H. acquired data and revised the manuscript; L.G.T.M. acquired and interpreted data and revised the manuscript; M.H. acquired data and revised the manuscript; UKeller interpreted data and revised the manuscript; T.C. acquired and analyzed data; H.G. and S.F. acquired data and funding, designed the work and revised the manuscript; S.O. acquired and interpreted data and revised the manuscript; U Keilholz designed the work, interpreted data and revised the manuscript; E.B., D.B. and D.T.R. contributed to the conception and design of the work, interpretation and acquisition of data, acquisition of funding and drafted and revised the manuscript.

## Funding

## Competing interests

C.H. received honoraria, research funding and/or consulting/advisory board from Roche, Novartis, and Boehringer Ingelheim. S.F. reports consultancy fees from Illumina, DTR has received honoraria, research support, and/or travel/accommodation expenses from Bayer, Eli Lilly, Bristol-Myers Squibb, Roche, BeiGene, J&J, and Seagen. The remaining authors report no competing interests.

## Additional information

[1]Core Unit Bioinformatics, Berlin Institute of Health at Charité–Universitätsmedizin Berlin, Berlin, Germany. [2]Department of Hematology, Oncology and Cancer Immunology, Campus Benjamin Franklin, Charité–Universitätsmedizin Berlin, Corporate Member of Freie Universität Berlin and Humboldt-Universität zu Berlin, Berlin, Germany. [3]Department of Pathology, Charité-Universitätsmedizin Berlin, Corporate Member of Freie Universität Berlin and Humboldt-Universität zu Berlin, Berlin, Germany. [4]Comprehensive Cancer Center, Charité - Universitätsmedizin Berlin, Corporate Member of Freie Universität Berlin and Humboldt-Universität zu Berlin, Berlin, Germany. [5]German Cancer Consortium (DKTK), partner site Berlin, a partnership between DKFZ and Charité - Universitätsmedizin Berlin, Berlin, Germany. [6]National Center for Tumor Diseases (NCT), NCT Berlin, a partnership between DKFZ, Charité Universitätsmedizin, BIH and MDC, Berlin, Germany. [7]German Cancer Consortium (DKTK), DKFZ, core center Heidelberg, Heidelberg, Germany. [8]Institute of Pathology, Ludwig-Maximilians-University Munich, München, Germany. [9]Division of Translational Medical Oncology, German Cancer Research Center (DKFZ), Heidelberg, Germany. [10]National Center for Tumor Diseases (NCT), NCT Heidelberg, a partnership between DKFZ and Heidelberg University Hospital, Heidelberg, Germany. [11]Department for Translational Medical Oncology, National Center for Tumor Diseases Dresden (NCT/UCC), a partnership between DKFZ, Faculty of Medicine and University Hospital Carl Gustav Carus, TUD Dresden University of Technology, and Helmholtz-Zentrum Dresden - Rossendorf (HZDR), Dresden, Germany. [12]Translational Medical Oncology, Faculty of Medicine and University Hospital Carl Gustav Carus, TUD Dresden University of Technology, Dresden, Germany. [13]German Cancer Consortium (DKTK), partner site Dresden, Dresden, Germany. [14]German Cancer Consortium (DKTK), partner site Munich, Munich, Germany. [15]West German Cancer Center, University Hospital Essen Essen, Germany. [16]German Cancer Consortium (DKTK), partner site Essen, Essen, Germany. [17]Institute of Medical Bioinformatics and Systems Medicine, Medical Center-University of Freiburg, Faculty of Medicine, University of Freiburg, Freiburg, Germany. [18]German Cancer Consortium (DKTK), Partner site Freiburg, a partnership between DKFZ and Medical Center - University of Freiburg, Freiburg, Germany. [19]Goethe University Frankfurt, University Hospital, Department of Medicine, Hematology/Oncology & University Cancer Center Frankfurt (UCT), Frankfurt, Germany. [20]German Cancer Consortium (DKTK), partner site Frankfurt, Frankfurt, Germany. [21]Department of Surgery, Memorial Sloan Kettering Cancer Center, New York, NY, USA. [22]Department of Oral and Maxillofacial Surgery, Charité - Universitätsmedizin Berlin, Corporate Member of Freie Universität Berlin, Humboldt-Universität zu Berlin, Berlin, Germany. [23]Genomics Technology Platform, Berlin Institute of Health at Charité - Universitätsmedizin Berlin and Max Delbrück Center for Molecular Medicine in the Helmholtz Association, Berlin, Germany. [24]Institute of Human Genetics, Heidelberg University, Heidelberg, Germany. [25]These authors contributed equally: Eric Blanc, Dieter Beule, Damian T. Rieke. ✉e-mail: damian.rieke@charite.de

