## [Transparent Peer Review file · Nature Communications]

A macrophage-predominant immunosuppressive microenvironment and therapeutic vulnerabilities in advanced salivary gland cancer

Corresponding Author: Dr Damian Rieke

A version of this paper was originally rejected for publication by Nature Communications, however that decision was reconsidered after appeal by the authors.

Version 1:

Reviewer comments:

Reviewer #1

(Remarks to the Author)

This study utilizes a wide range of bioinformatics approaches and, through computational prediction, reveals the complexity of the immune microenvironment in SGC, highlighting its subtype-specific characteristics. The authors found that the majority of advanced SGC samples, particularly those in ACC subgroup 1, exhibit an immune-deserted profile with downregulation of the antigen processing machinery. The single-cell and bulk analyses indicate an M2 macrophage-dominant TIM across SGC subtypes. These findings suggest the presence of an immunosuppressive TIM in SGC and point to subtype-specific vulnerabilities to immunotherapeutic strategies. The observation that a high T-cell to macrophage ratio is associated with a response to immune checkpoint inhibitors (ICI) in individual patients could have important clinical implications. The identification of high VTCN1 (B7-H4) expression in ACC is noteworthy, as it suggests potential new therapeutic targets. On the other hand, there are several aspects that require improvement and some concerns that need to be addressed. Resolving these issues would be critical to enhance the study's impact and credibility.

Major concerns:

1. The claim of a potential biomarker lacks persuasiveness due to the insufficient number of responders. Although validated by another cohort, the results remain inconclusive because of the small sample size. It is recommended to expand the external validation set and moderate claims about biomarkers or therapeutic targets. While SGC is a rare cancer with low response rates to immune checkpoint inhibitors, drawing conclusions based on a single responder is inadequate. In the validation set, there is only one responder, and another patient with a similar T cell fraction is a non-responder. Given the current results, it's challenging to accept the conclusion that the highest T cell fraction indicates a high likelihood of response.
2. Only a small portion of cell type fractions were calculated in the study. For example, while the macrophage fraction was estimated in the results (figures 3-5), it's unclear how the M2 polarized macrophage fraction was measured. It's suggested to use immune cell deconvolution tools for more comprehensive cell typing and to consider the M1/M2 ratio. Additionally, it needs to be clearly demonstrated that the T-cell/macrophage ratio is actually a T-cell/M2 ratio, and the current results lack sufficient evidence (sample size, rationale, etc.) to show how this ratio is involved in immune modulation.
3. To enhance result reliability, it's suggested to add figures for some results where only statistical values are described. For instance, the association between tumor purity and inflammation, and the correlation between IFNG score and immune cell proportion are mentioned in the text but lack visual representation.
4. There are concerns about the study's novelty, dataset, and abundance of evidence. Recent studies have explored tumor microenvironments across salivary gland cancer subtypes, and the reduced immunity in ACC has been explained in several papers. The differences from existing research need to be emphasized.

Minor concerns:

1. Throughout the paper, the size of figures and font sizes are too small and unrefined, making them difficult to view. Improvement is needed. Particularly, the wordcloud size in Figure S1A is too small to see. Also, a detailed explanation of the y-axis in Figure S1B is needed in the figure caption.
2. It's suggested to describe the AUC calculation method for Figure S1B.
3. In Fig 2, many signatures are used, but clearer explanations are needed. For example, the meaning of "2 different sets of signatures of immune infiltrates" is ambiguous.
4. The text in Figure S2B should be converted to bold.
5. The abbreviations for ID and ED are missing.
6. An objective threshold for medium to strong intensity in IHC is needed.
7. In Fig 3A, P3 is missing the immune phenotype due to unavailable RNA-seq data. It's suggested to add NA to the legend.
8. Among the 6 major cell types in Fig 3 A-B, alveolar cells are only considered in lung metastasis, but there's a possibility that samples obtained from primary tumors could be typed as these alveolar cells.
9. The statistical indications in Fig 4B are unclear. For example, the NS, presumably for Immune-high and medium, is confused with other indications.
10. There are no results for the correlation between bulk RNA expression and IHC results in Figure 5D-E.
11. Results showing that ACC1 has higher VTCN1 expression than ACC2 samples are needed in Figure 5G-F.
12. On page 7, in line 5, replace 'withGSVA' with 'with GSVA'.

Reviewer #2

(Remarks to the Author)

This manuscript details the results of a multiomic evaluation of various cancer tumors of the salivary gland in a cohort of 104 subjects that presented with a variety of different salivary gland cancers. The manuscript details the results of bulk RNA-sequencing available n=93 samples used to identify altered transcriptional profiles in each subtype. In addition, the all subjects have either WES (=53) or WGS (n=50) that was used to determine the role of somatic mutation in subsets salivary gland cancer. Lastly, some single nuclei RNAseq data is available and was used to help refine the cell types the transcriptional changes were likely to impact in the bulk RNAseq dataset. Some of these findings are supported by additional immunohistochemical images. The major findings focus not the role of the immune dysregulation observed in the bulk RNAseq dataset that shows a role in ACC for down regulation of antigen processing and some ACC showed T cell inflammation pathways. Immune checkpoints and cancer testis antigen was shown to coincide with VTCN1 expression with in ACC. Once integrating the snRNAseq and bulk RNAseq, M2 macrophages dominated the tumor microenvironment with high T cell to macrophages ratio patients showing the most clinical benefit of checkpoint inhibitor therapies. While this manuscript converts an important topic and presents some interesting findings, there are several issues that need to be addressed. Most importantly the organization throughout the manuscript makes it very difficult to follow. Moreover, some of the panels are not organized appropriately which further increases the difficulty with this manuscript. Until the manuscript is substantially revised, it is difficult to determine the rigor of this work.

- 1). Some of the sample sizes for the various cancers are very small and should be omitted from the majority of the manuscript. Most of the finding focus on the ACC subjects and these could be brought as comparator groups, but it is not possible to draw conclusions from such small numbers.
- 2). This manuscript needs a summary figure that is more descriptive and provides more description of how the analyses was performed throughout the study. While figure 1 provides some framework it does not lets us see how these were used in combination to draw the conclusions presented in the manuscript.
- 3). Table 1 is very difficult to read and does not allow for us to see how each category is related to each other. This table needs to be revised and shown so that we can see the distribution of each category across each tumor type. The table also does not present ancestry.
- 4). There are several places where the figures are not reference correctly. For example, Figure 5B states that it is validation of results from Figure E. Also Figure 2B is referenced in the text before Figure 2A with Figure 2F reference next.
- 5). The methods are split up such that related items are not sequential. Example is that sequencing of any kind has been lumped into one section (Exome, Genome, and RNA sequencing) which then leaves the WGS and WES analytical approaches many paragraphs separated from the sequencing methods used.
- 6). Only 20X converse n the mutant allele is very shallow. How variants of interest were identified is not presented in the methods.
- 7). Figure 2 should omit panel A, C, and D from the main text and move to the supplement since they are not the primary focus of the results presented and make the panel hard to discuss in the text in order.
- 8). Was the results of Figure 3 C also clustered as major immune subpopulations or only together as presented in the text?
- 9). Since so many of the approaches have been done in the same subject, it is not validation of the finding. The authors need to be more clear on how they subdivided their data to make this more clear to the authors.

10). There are many limitations to this study yet none are discussed.

Reviewer #3

(Remarks to the Author)

Overall, the manuscript is interesting and the datasets are important. Salivary gland malignancies are understudied and the authors attempt to make an important advance in understanding inflammation and its relationship to therapeutic response in tumors types without many therapeutic options outside of standard chemo and surgical excision. Thus there is a critical need for these datasets to be shared widely in a public repository so that others can verify and learn from these types of data. After my initial reading and enthusiasm for the manuscript, it is clear that the authors have overlooked important analyses that are required for publication. Moreover, the structure of the manuscript, the thinness of the experimental details, incompleteness of the figures, and lack of data availability attenuated the enthusiasm of the manuscript.

This reviewer does not have access to the sequencing data to evaluate its veracity and interpretation. Independent review of the sequencing data should be performed by an informaticist.

The paper requires editing for punctuation and clarity.

The figures, while neatly constructed, lack sufficient details and legends to adequately interpret the data (e.g., Figure 5). Some supplemental findings are very important and could be included in main figures, which could be a bit more robust (e.g. Figure 4, 5 and associated subs). Overall, some legend details are lacking.

General comments on the main findings:

Given the size and depth of molecular characterization of the collected tumors, the author's decision not include patient-level mutation data for cancer-associated mutations/translocations/CNV in each SGC case (e.g., TP53, PIK3CA, PTEN, HRAS, KRAS,...) is a major limitation.

M2 macrophages in the salivary glands are also the predominant macrophages in normal immune infiltrates in the salivary glands - so-called "muciphages", mucin-laden macrophages that engulf spilled mucins in the extracellular space. Mucin spillage is very common at the advancing edge of tumors in the salivary glands given compression and disruption of the ductal network. Some comparison to normal age-matched salivary glands' M2 macrophage phenotypes. This may include adjacent normal nuclei from FFPE blocks of the tumors for snRNAseq and IF/ISH mapping of M2 m-phage at the tumor-parenchymal interface.

There are far too many examples to regurgitate here, but the authors failure to provide the full names of abbreviations limits interpretability e.g., TAM (tumor associated macrophage?), ED/ID-histologies?,

The authors miss a crucial opportunity to report/understand in SGC the relationship b/w histological type, translocation status, and TMB.

Specific Comments by section

Results

Cohort characteristics

In addition to Table 1, a supplemental table 1 that shows de-identified coded patient-level metadata on the type of tumor, stage, age, sex, presences of high grade features or HGT, treatment, and analysis modalities on all cases/controls. The current table, while useful for summary statistics, is insufficient. "A summary of molecular data layers is provided in Figure 1." is inadequate and does not permit the reviewers from understanding which cases were used for various 'omics modalities.

The authors provide validation of translocation status in AdCC (adenoid cystic carcinoma), it is necessary to provide the same for the other cases where the authors have data modalities that permit derivation of this data (e.g., WGS, bulk RNAseq, WES). Provide said data in the suggested Sup Table 1. Report any molecularly-based reclassification (e.g., misdiagnosed pleomorphic adenocarcinoma, nee PLGA, PA, etc). There are informatics pipelines that can be run to identify known translocations on these datasets. Furthermore, Acinic Cell Carcinoma (AcCC), with translocation representing enhancer hijacking to the promoter of the TAD of SCPP gene cluster can be derived from NR3A3 GEX (it will be 10-fold higher in the bulk RNAseq space).

"Around 65% of ACC had a MYB-NFIB and 4% a MYBL1-NFIB fusion, in concordance with the reported proportions from literature. None of the non-ACC samples had a MYB-NFIB fusion." I think that with strong histopathology, the rate of positivity in fresh samples is higher than frozen (~85 vs 44%). What other mutations/translocations were identified for these cases? While this is within the range of what is reported the literature, stating "concordance" is not supported. The authors have the capacity to confirm these translocations using their multiomics platform (or molecular reclassification), as stated in this manuscript. Comprehensive characterization of the mutations using their 'omics data is required (WGS will pick up tumor-

associated mutations, deletions, CNV, etc; WES will pick up translocations involving coding regions - e.g., MYB-NFIB, MECT1-MAML2

"PC1, which separated ACC from other entities, was related to immune response (Figure S1A)." Please provide a figure within this sup figure of the PC1 loadings (what genes specifically drive this difference).

"A primary analysis of TIM composition was performed using scores for 2 different sets of signatures of immune infiltrates." Contextualize how these indices were selected? Provide references or how the authors derived this choice.

"Yet, we identified a small subset of ACC samples (n=10, ~17%) with high immune infiltration, which could also be validated in a separate analysis in only ACC samples (n=57)." This requires further analysis using the data the authors already have generated. The authors shall consider the correlation of translocation status with TMB, after assigning driver mutations amongst the available cases (as requested above). In other studies, TMB is higher in non-translocation positive tumors (e.g., salivary duct carcinoma, epithelial-myoepithelial carcinoma with HGT). Do these high mutation burden samples lack translocations? Alternately, are these high inflammation also HGT samples? These details all require further reporting.

Tumor mutational burden description - "Mutational signatures 48 were extracted for WGS data. Four main signatures could be found in 52 patients with WGS including APOBEC-signatures (SBS2,13), chemotherapy." Where is this data presented in the manuscript? mutational burden signature enrichment in each case should be included in a table linked to the anonymous identifiers used in the patient-level sup table 1. Report known driver mutations in all samples c/w TCGA top.

"Since TMB was not associated with inflammation in ACC samples..." Where is this data or show the relationship amongst the two.

Single nuclei sequencing:

The cohort is very small and heterogenous to the point that meaningful associations and conclusions cannot be drawn outside of the 5 cases of AdCC. While 5 AdCC cases were sequenced (this is in my understanding the first published scRNAseq of salivary gland tumors), how many of these cases had high and low inflammation burden? An expanded cohort of snRNAseq of additional cases or using FFPE blocks using the 10X Flex kit would vastly improve the generalizability of this data. Please explain how it is possible that not a single normal epithelial cell (duct, acini?!) is represented across 85,000 cells from primary tumors from the glands WHEN the parent block has advancing tumor - tumor stroma - parenchymal interface. Supplemental data or supplemental methods should show how/where nuclei were selected from parent frozen blocks for single nuclei. MOREOVER, showing the expression of target genes in the translocation positive cells along with VCTN1 would be interesting since it is well appreciated in the biphasic tumor that the translocation can be positive in both luminal and abluminal cells.

Please separate IgG and IgA plasma cells into two clusters by manual annotation.

Figure 3A requires case codes for each stacked bar chart.

Fig3C. Show M1 and M2 since you discuss M2 polarization. Show dot-plots showing expression differences and markers for M1 and M2 in the figure.

Please explain how immune low, medium, and high cutoff were established and how a "low vs "high" might change the findings? It seems that medium and high almost intersect in your data whereas low seems to be distinct.

Figure 4E needs annotation (dotted outline) of the tumor or use of another color. Representative LOW and HIGH CD68 infiltration is needed. M2 vs M1 marker to confirm M2 polarization. Mucicarmin or PAS to evaluate if they are mucin laden. Figure 4E would be aided by IF/IHC showing CD4/CD8 T cells and/or CD38/CD138+ Plasma cells in the tumor stroma. Figure 4D should add "immune medium line" since it is reported in the preceding figures.

"TIM of advanced SGC to be Macrophage-dominated" The authors do not provide a definition of what an advanced SGC is versus any other carcinoma. There is no patient level data table that references this data.

"On the other hand, no significant effect of macrophage proportion was found on overall survival, although a tendency to shorter survival in samples with high macrophage content was observed (data not shown)." This data is important and consistent with other data in the manuscript. It is inappropriate to withhold it. Please include.

"Abundance of macrophages and T-cells were correlated inversely both in bulk data ($\rho=-0.57, p=5e-09^{***}$) as well as in single nuclei data ($\rho=-0.91, p=1e-05^{***}$)." Mapping of M-phage and T cell cases within one tumor type stratified by tumor stage or high grade features, or something is needed here. Please show evidence that this relates to tumor parameters/tumor type/tumor grade, etc.

Also, the majority of macrophages were predicted as being M2 polarized (median 81% of all macrophages). Both points indicate an immune-suppressive role of macrophages in the TIM." While this is well-understood for other cancers, the authors overlook an important aspect of the salivary glands. M2 macrophages in the salivary glands cluster with the predominant macrophages in normal resident immune infiltrates in the salivary glands - so-called "muciphages", mucin-laden macrophages that engulf spilled mucins in the extracellular space. Mucin spillage is common at the advancing edge of tumors in the salivary glands given compression and disruption of the ductal network, whether or not they are immune suppressive. Some comparison to normal age-matched salivary glands' M2 macrophage phenotypes is needed to make this

claim above and to show how they are phenotypically different AND immune suppressive in the SGC tumor context. An alternate explanation is that, the M2 may not be immune suppressive as stated, but merely a consequence of the tumor on the parent organ causing mucin spillage. This may include adjacent normal nuclei from FFPE blocks of the tumors for snRNAseq and IF/ISH mapping of M2 m-phage at the tumor-parenchymal interface.

Figure 5: What is the top red/mauve/pink column at the tops of the stacked bar chart? These types of omitted details are concerning. 5H violin plot. Figure 5C shows the immune clustering clusters more naturally into two clusters, not three. This could be discussed further as previously identified.

Report TMB for the ICI-treated responders/non-responders. My hypothesis is the adenocarcinoma NOS (not really an appropriate molecularly-based diagnosis without exhaustive molecular investigation showing that it is NOT a known salivary gland malignancy - especially in the face where the authors can report the mutation status of this case based on available "multiomics" data). It is curious that the two tumors responding to ICI are classes enriched for high mutational burden (Salivary duct carcinoma and high grade adenocarcinomas that are difficult to classify without appropriate ancillary testing). This is an important caveat to report.

Where is the mention on what the two groups of ACC tumors are i.e., ACC1 and ACC2 groups?

Discussion:

"Inflammation was associated with tumor mutational burden mostly in non-ACC rather than ACC histologies in our analyses, albeit the effects were marginal which might be related to an overall lower tumor mutational burden in the majority of samples" but why - please elaborate.

"The high macrophage content, in particular of M2-polarized macrophages in advanced salivary gland cancers, might therefore mediate immune checkpoint inhibitor resistance in these patients despite the presence of an inflamed TIM." Some experimental validation of this in AdCC would help make this point better.

Version 2:

Reviewer comments:

Reviewer #1

(Remarks to the Author)

The authors have effectively addressed our concerns by collecting and validating data across multiple levels of omics. However, the T-cell/macrophage ratio remains a point of concern as a potential biomarker for clinical benefit. While they noted that clinical benefits were observed only in patients with a high T-cell/macrophage ratio, the limited number of such patients and the ambiguity surrounding the criteria for defining this ratio raise important questions. For instance, it appears that there are more patients who, despite having a T-cell/macrophage ratio similar to those who respond, do not exhibit any clinical responses. Therefore, uncertainty remains about whether a high T-cell proportion can reliably serve as a biomarker for immunotherapy. Please elaborate on this concern.

Reviewer #2

(Remarks to the Author)

No further Comments.

Reviewer #3

(Remarks to the Author)

The authors have addressed very nearly all the reviewers' comments. Their revised manuscript has improved the manuscript in multiple ways. Most importantly, the logic of the manuscript is greatly improved.

One persistent issue is the lack of consistent and systematic organization of the figures. While a minor issue scientifically, it is disruptive to the interpretation within the scientific context of the manuscript (requires going back/forth b/w legends and text and both forwards and backwards within the figures). Some of the figures are not arranged alphabetically from left to right (e.g. B, then A). Within the same critique and similar to reviewers' comments in the previous version, the use of one legend for multiple different figures is challenging to visually (in an individual who does not have color blindness) link the legend to the figures.

Notably, the writing of the manuscript is also tremendously improved. However, there is a lack of coherence in writing between the new and the original sections. During editing, the authors may want to align the writing across the manuscript.

REVIEWER COMMENTS

Reviewer #1 (Remarks to the Author): computational expertise in multi-omics integration

Major concerns:

1. The claim of a potential biomarker lacks persuasiveness due to the insufficient number of responders. Although validated by another cohort, the results remain inconclusive because of the small sample size. It is recommended to expand the external validation set and moderate claims about biomarkers or therapeutic targets. While SGC is a rare cancer with low response rates to immune checkpoint inhibitors, drawing conclusions based on a single responder is inadequate. In the validation set, there is only one responder, and another patient with a similar T cell fraction is a non-responder. Given the current results, it's challenging to accept the conclusion that the highest T cell fraction indicates a high likelihood of response.

We acknowledge the reviewer's valid concern regarding the small sample size and the limited number of responders, which indeed makes it difficult to draw definitive conclusions about potential biomarkers. The lack of approved treatment options and clinical trials investigating immune checkpoint inhibitors in salivary gland cancer makes it difficult to expand the validation cohort.

Despite these limitations, we were able to expand our analysis, which now comprises:

Exploratory cohort: We identified and integrated an additional 6 SGC patients treated with immune checkpoint inhibitors after RNA-seq data analysis (now n=28 patients, clinical benefit in n=3). Clinical benefit was still observed in the three patients with non-ACC histology and highest T-cell/macrophage ratios, thus supporting prior findings. The updated figure and results of samples with significant results from deconvolution analysis are provided in Figure 6.

Validation cohort: In addition to available data from pre-treatment biopsies, we were able to retrieve post-treatment biopsy RNA-seq data.

After confirming that pre- and post-treatment biopsies did not impact the T-cell/macrophage ratio (paired t-test $p=0.62$), we separately analyzed post treatment biopsies (n=8, clinical benefit in 2) and again identified the highest T-cell/macrophage ratio in the two patients with clinical benefit.

Validation cohort 2: We additionally performed an immunohistochemical validation in a newly generated and independent cohort of 22 patients, of which 5 received immune checkpoint inhibition (one clinical benefit). Again, the highest T-cell/macrophage ratio was observed in the responding patient.

Our observations suggest that the TIM composition may act as an additional factor influencing the response to immune checkpoint inhibitors (ICIs). However, response rates to immune checkpoint inhibition are low. Furthermore we agree that response

to immune checkpoint inhibition is impacted by several factors, which is why we moderated claims both in abstract, results and discussion text.

2. Only a small portion of cell type fractions were calculated in the study. For example, while the macrophage fraction was estimated in the results (figures 3-5), it's unclear how the M2 polarized macrophage fraction was measured. It's suggested to use immune cell deconvolution tools for more comprehensive cell typing and to consider the M1/M2 ratio. Additionally, it needs to be clearly demonstrated that the T-cell/macrophage ratio is actually a T-cell/M2 ratio, and the current results lack sufficient evidence (sample size, rationale, etc.) to show how this ratio is involved in immune modulation.

We thank the reviewer for this important comment. We agree that the immune cell subtypes need to be considered. Following the reviewer's suggestion, we additionally utilized deconvolution tools (CIBERSORT) to predict the proportions of all T-cell subsets (including various phenotypes such as CD4+, CD8+, regulatory T cells, etc.) and all macrophage subsets (encompassing both M1 and M2 phenotypes) in bulk data. Here, the T-cell subset was almost equally consisting of CD4+ and CD8+ cells and the macrophage subset was dominated by M2 macrophages (~78%). We have specified these findings in the newly generated supplementary figures 8 and 9. We additionally harnessed single-cell data to perform an in-depth analysis of cell subtypes, which further confirmed the predominance of M2-macrophages (Suppl. Figure 8), thus demonstrating T-cell/M2 ratio. Following the reviewer's suggestion we also analyzed a ratio of different macrophage subsets, which indeed showed a slight difference in ACC compared to non-ACC samples (Figure 5).

3. To enhance result reliability, it's suggested to add figures for some results where only statistical values are described. For instance, the association between tumor purity and inflammation, and the correlation between IFNG score and immune cell proportion are mentioned in the text but lack visual representation.

We agree that visual representation is required to back up statistical findings. We have therefore added all plots/tables underlying statistical value to the manuscript. We added a general figure for all associations between clinical parameters and inflammation to Figure 2 and Suppl. Figure 5. Specifically we added the correlation plot of tumor purity vs IFNG to Suppl. Figure 5 and the IFNG vs Immune cell percentage (single cell) to Suppl. Figure 7.

4. There are concerns about the study's novelty, dataset, and abundance of evidence. Recent studies have explored tumor microenvironments across salivary gland cancer subtypes, and the reduced immunity in ACC has been explained in several papers. The differences from existing research need to be emphasized.

We acknowledge that recent research has explored the tumor microenvironment (TIM) across various salivary gland cancer (SGC) subtypes and have referenced

previous work. However, we believe our study offers several novel contributions that differentiate it from existing research:

- Our cohort includes rare SGC subtypes, which are not represented in the majority of available SGC studies.
- Harnessing multi-omics data layers allows for novel insights such as VTCN1 expression heterogeneity within ACC subtypes and TIM composition.
- This study provides a multi-omics resource including WES/WGS/RNA-Seq and single-nuclei RNA sequencing (scRNA-seq) as well as clinical data.
- Our study comprises only patients with recurrent and metastatic cancers, thus representing an intention-to-treat cohort for systemic therapies including immunotherapies. Previous analyses are focusing mainly on localized/primary tumors.
- We have contextualized the TIM composition by comparing it with healthy salivary gland tissue, other cancer types and previously published SGC cohorts.

Following the reviewer's suggestions, we have highlighted these factors in the manuscript.

Minor concerns:

1. Throughout the paper, the size of figures and font sizes are too small and unrefined, making them difficult to view. Improvement is needed. Particularly, the wordcloud size in Figure S1A is too small to see. Also, a detailed explanation of the y-axis in Figure S1B is needed in the figure caption.

We thank the reviewer for this feedback. We have extensively revised, improved and re-structured all figures to improve readability.

2. It's suggested to describe the AUC calculation method for Figure S1B.

We have added the description in the methods.

3. In Fig 2, many signatures are used, but clearer explanations are needed. For example, the meaning of "2 different sets of signatures of immune infiltrates" is ambiguous.

We agree that this was difficult to understand and have therefore revised methods and results sections to better explain the signatures.

4. The text in Figure S2B should be converted to bold.

We changed the structure of the figure. The plot mentioned is now found in FigS3B with larger fonts.

5. The abbreviations for ID and ED are missing.

We have added the abbreviation in the manuscript text and in the abbreviation list. We have additionally double-checked the manuscript to make sure all abbreviations are introduced.

6. An objective threshold for medium to strong intensity in IHC is needed.

We have added definitions and thresholds to the method section.

7. In Fig 3A, P3 is missing the immune phenotype due to unavailable RNA-seq data. It's suggested to add NA to the legend.

We have updated the figure to include a legend for NAs.

8. Among the 6 major cell types in Fig 3 A-B, alveolar cells are only considered in lung metastasis, but there's a possibility that samples obtained from primary tumors could be typed as these alveolar cells.

In order to rule out mis-labeling, we double-checked single-cell data and did not identify any cell labelled as alveolar cell in any sample from primary tissue or other metastatic sites (lymph node/skin).

9. The statistical indications in Fig 4B are unclear. For example, the NS, presumably for Immune-high and medium, is confused with other indications.

We thank the reviewer for pointing this out and have changed the figure to clearly label statistical implications of immune-group differences.

10. There are no results for the correlation between bulk RNA expression and IHC results in Figure 5D-E.

We added the p-value and changed the plot accordingly (Fig 7D).

11. Results showing that ACC1 has higher VTCN1 expression than ACC2 samples are needed in Figure 5G-F.

Following the reviewer's suggestion, we have made sure that all results now have visual representation. The requested plot has been added to Figure 7B.

12. On page 7, in line 5, replace 'withGSVA' with 'with GSVA'.

Thank you for pointing out this typo. We have corrected this.

Reviewer #2 (Remarks to the Author): expertise in salivary gland immune microenvironment

1). Some of the sample sizes for the various cancers are very small and should be omitted from the majority of the manuscript. Most of the finding focus on the ACC subjects and these could be brought as comparator groups, but it is not possible to draw conclusions from such small numbers.

We thank the reviewer for this important comment. We agree, that the heterogeneity of the cohort is both a strength and limitation. It is a strength because we here

provide data on very rare and understudied subtypes. Yet, we agree that conclusions can not be drawn for subtypes with only limited numbers. We agree, that the most robust findings are from ACC and non-ACC samples serve mainly as comparators and have now highlighted these limitations in the discussion section.

Since salivary gland cancer subtypes harbor biological similarities within excretory-duct-like (ED) and intercalated-duct-like (ID) groups, we have additionally included these subtypes, thus allowing for an improved generalizability of results. In a combined analysis of available published datasets a sufficient number of samples (n>20) was only evaluable for ACC, myoepithelial carcinoma and salivary duct carcinoma and revealed significant differences between the three tumor types (Suppl. Figure 4D).

2). This manuscript needs a summary figure that is more descriptive and provides more description of how the analyses was performed throughout the study. While figure 1 provides some framework it does not let us see how these were used in combination to draw the conclusions presented in the manuscript.

We agree, that the multiple data layers are difficult to follow. Following the reviewer's suggestion, we have added a new summarizing figure showing all data types and analyses (Suppl. Figure 1).

3). Table 1 is very difficult to read and does not allow for us to see how each category is related to each other. This table needs to be revised and shown so that we can see the distribution of each category across each tumor type. The table also does not present ancestry.

We thank the reviewer for this suggestion. We have changed table 1 to allow for a separate view of ACC and non-ACC samples. We additionally included a newly generated supplemental table 1 to allow for an in-depth look at individual characteristics.

Unfortunately, ancestry data are not collected within the DKTK MASTER program but the vast majority of patients have a central European genetic background. We have added this information to the manuscript results.

4). There are several places where the figures are not reference correctly. For example, Figure 5B states that it is validation of results from Figure E. Also Figure 2B is referenced in the text before Figure 2A with Figure 2F reference next.

We apologize for these inconsistencies. We have double-checked the manuscript to make sure all figures are referenced correctly.

5). The methods are split up such that related items are not sequential. Example is that sequencing of any kind has been lumped into one section (Exome, Genome, and RNA sequencing) which then leaves the WGS and WES analytical approaches many paragraphs separated from the sequencing methods used.

We agree, that the combined description makes understanding the methods difficult. However, the integrated and comprehensive nature of performed analyses also makes it difficult to describe them in a sequential fashion without losing

brevity. In order to allow for a brief but comprehensive understanding of methods, we have revised the methods section. We now describe the analytical approaches under the following subheadings:

- Immunohistochemistry
- WES/WGS/RNA-Sequencing
- Sequencing data processing
- Bulk RNASeq downstream analysis
- Clinicogenomic integrated analysis
- WES/WGS sequencing data downstream analysis
- Single-nuclei sequencing
- Single-nuclei sequencing data processing and downstream analysis

6). Only 20X coverage in the mutant allele is very shallow. How variants of interest were identified is not presented in the methods.

We apologize for not adequately presenting methods used. Variants were called using Mutect2. Mutect2 inherently performs stringent filtering by removing sequencing artifacts, addressing orientation bias, and applying probabilistic models to distinguish true somatic mutations from noise, while handling low-coverage regions through dynamic error modeling, minimal coverage thresholds, and integration with germline and normal panel databases to ensure high-confidence variant calls.

We adjusted the threshold for coverage of the mutant allele to 50x for the TMB calculation in WES data. This was not possible to do in WGS data as it has an overall lower coverage. These information have been updated in the methods part.

7). Figure 2 should omit panel A, C, and D from the main text and move to the supplement since they are not the primary focus of the results presented and make the panel hard to discuss in the text in order.

We agree that figure 2 was difficult to interpret and have changed it according to the reviewer's comments. Panel A, C and D were moved to newly generated Suppl. Figure 4.

8). Was the results of Figure 3 C also clustered as major immune subpopulations or only together as presented in the text?

We initially analyzed the TIM including major immune subpopulations, which is shown in Figure 3C and D. We agree that the graphical depiction of immune cell subpopulations is essential, which is why we have included 2 newly generated figures showing specific cell subpopulations for macrophages (Suppl. Figure 8) and T-cells (Suppl. Figure 9) .

9). Since so many of the approaches have been done in the same subject, it is not validation of the finding. The authors need to be more clear on how they subdivided their data to make this more clear to the authors.

We thank the reviewer for pointing out the important issue of validation. We agree, that internal validation using different techniques is not sufficient to validate

findings. Exploratory analyses were performed in the main dataset integrating WES/WGS, Transcriptome, single-nuclei sequencing and immunohistochemistry. This dataset was expanded by an additional 6 patients treated with immune checkpoint inhibitors (Figure 6). Validation was performed in external datasets (Vos et al pretreatment data, Vos et al. posttreatment data, Linxweiler et al., TCGA data). Following the reviewer's suggestions, an additional, external dataset of immunohistochemistry data was generated (n=22).

10). There are many limitations to this study yet none are discussed.

We agree, that many limitations exist for this study. We acknowledge that addressing the study's limitations is crucial for providing context to our findings and for guiding future research.

The primary limitations of our study are:

- 1. Low Sample Size:** Due to the rarity of advanced-stage salivary gland carcinomas (SGCs), our cohort includes a relatively small number of samples given the high dimensionality of sequencing data and inclusion of rare subtypes. This limited sample size reduces the statistical power of our analyses and may affect the generalizability of our results. We recognize that conclusions drawn from a small cohort should be interpreted with caution and considered preliminary.
- 2. High Heterogeneity:** Our cohort is highly heterogeneous, encompassing a wide range of SGC subtypes, biopsy sites, and treatment histories. While this diversity allows us to capture a broader spectrum of tumor biology, it also introduces variability that can confound the interpretation of our data. The heterogeneity makes it challenging to attribute specific findings to particular tumor types or to make overarching conclusions applicable to all SGCs.
- 3. Multiple Covariates Influencing Results:** Several covariates, such as differences in prior treatments, may influence our results. The interplay of these factors with the TIM composition can complicate the analysis and interpretation of the data, potentially masking or mimicking significant associations.

We have highlighted these points in the discussion section.

Reviewer #3 (Remarks to the Author): expertise in salivary gland tumours

This reviewer does not have access to the sequencing data to evaluate its veracity and interpretation. Independent review of the sequencing data should be performed by an informaticist.

We have uploaded sequencing data (WES/WGS/Transcriptome) to the The European Genome-phenome Archive (EGA) under the following identifier: EGAS50000000809. Single-nuclei data are also being uploaded to the The European Genome-phenome Archive (EGA) and will be provided as soon as the upload has been completed.

The paper requires editing for punctuation and clarity.

We have extensively edited the manuscript to improve punctuation and clarity.

The figures, while neatly constructed, lack sufficient details and legends to adequately interpret the data (e.g., Figure 5). Some supplemental findings are very important and could be included in main figures, which could be a bit more robust (e.g. Figure 4, 5 and associated subs). Overall, some legend details are lacking.

We thank the reviewer for this favorable assessment. We have revised all figures to include relevant details and have furthermore included new figures and supplemental material to support the required details.

General comments on the main findings:

Given the size and depth of molecular characterization of the collected tumors, the author's decision not include patient-level mutation data for cancer-associated mutations/translocations/CNV in each SGC case (e.g., TP53, PIK3CA, PTEN, HRAS, KRAS,...) is a major limitation.

We agree, that the comprehensive data allow for additional in-depth analysis. We have so far limited some of these analyses to limit the number of covariates but agree, that the integration of patient-level mutation data is essential. Following the reviewer's suggestions, we have now integrated cancer-associated molecular alterations in the newly generated Figure 2 and Suppl. Table 1.

M2 macrophages in the salivary glands are also the predominant macrophages in normal immune infiltrates in the salivary glands - so-called "muciphages", mucin-laden macrophages that engulf spilled mucins in the extracellular space. Mucin spillage is very common at the advancing edge of tumors in the salivary glands given compression and disruption of the ductal network. Some comparison to normal age-matched salivary glands' M2 macrophage phenotypes. This may include adjacent normal nuclei from FFPE blocks of the tumors for snRNAseq and IF/ISH mapping of M2 m-phage at the tumor-parenchymal interface.

We thank the reviewer for this interesting comment. We agree, that the predominance of macrophages in salivary gland cancer might be related to physiological findings. We have therefore analyzed available bulk- and single-cell sequencing data of healthy salivary glands. We identified striking differences in the immune microenvironment between healthy and diseased tissue and especially identified different macrophage phenotypes (newly generated Suppl. Figure 12). In order to rule out the presence of muciphages as a cause of macrophage predominance n=34 samples were evaluable for PAS staining. Only one single PAS+ macrophage was identified in one sample (Suppl. Figure 10). We furthermore performed an in-depth analysis of tumor-associated macrophages in SGC (Suppl. Figure 8).

There are far too many examples to regurgitate here, but the authors failure to provide the full names of abbreviations limits interpretability e.g., TAM (tumor associated macrophage?), ED/ID-histologies?,

We apologize for our failure to provide sufficient explanations. We have carefully

revised the manuscript to make sure that all abbreviations are introduced.

The authors miss a crucial opportunity to report/understand in SGC the relationship b/w histological type, translocation status, and TMB.

We agree that histological subtypes including translocation status and TMB are important. Following the reviewer's suggestion we included a comprehensive assessment of molecular alterations, which are summarized in figure 2. We specifically focused on histological types and translocation status and have added these considerations to the results section. We additionally added TMB data to figure 2 to allow for an assessment of relationships in SGC and identified an overall lower TMB in translocation-associated SGC.

Specific Comments by section

Results

Cohort characteristics

In addition to Table 1, a supplemental table 1 that shows de-identified coded patient-level metadata on the type of tumor, stage, age, sex, presences of high grade features or HGT, treatment, and analysis modalities on all cases/controls. The current table, while useful for summary statistics, is insufficient. "A summary of molecular data layers is provided in Figure 1." is inadequate and does not permit the reviewers from understanding which cases were used for various 'omics modalities.

We agree, that the provision of in-depth patient-level data is important and have provided all available data with the manuscript (newly generated Suppl. Table 1)

The authors provide validation of translocation status in AdCC (adenoid cystic carcinoma), it is necessary to provide the same for the other cases where the authors have data modalities that permit derivation of this data (e.g., WGS, bulk RNAseq, WES). Provide said data in the suggested Sup Table 1. Report any molecularly-based reclassification (e.g., misdiagnosed pleomorphic adenocarcinoma, nee PLGA, PA, etc). There are informatics pipelines that can be run to identify known translocations on these datasets. Furthermore, Acinic Cell Carcinoma (AcCC), with translocation representing enhancer hijacking to the promoter of the TAD of SCPP gene cluster can be derived from NR3A3 GEX (it will be 10-fold higher in the bulk RNAseq space).

We agree, that molecular characteristics are essential for SGC diagnoses. Following the reviewer's suggestion we performed a molecular-based reclassification of samples. Indeed, a single PLAG1-fusion (consistent with carcinoma ex pleomorphic adenoma) positive sample in our cohort was initially diagnosed as an adenoid cystic carcinoma and another EWSR-fusion positive sample (consistent with clear-cell carcinoma) as adenocarcinoma NOS. Another adenocarcinoma NOS harbored high-level HER2 amplification (consistent with Salivary duct carcinoma). We have added these findings to the results section.

"Around 65% of ACC had a MYB-NFIB and 4% a MYBL1-NFIB fusion, in concordance with the reported proportions from literature. None of the non-ACC samples had a MYB-NFIB fusion." I think that with strong histopathology, the rate of positivity in fresh samples is higher than frozen (~85 vs 44%). What other mutations/translocations were

identified for these cases? While this is within the range of what is reported the literature, stating "concordance" is not supported. The authors have the capacity to confirm these translocations using their multi omics platform (or molecular reclassification), as stated in this manuscript. Comprehensive characterization of the mutations using their 'omics data is required (WGS will pick up tumor-associated mutations, deletions, CNV, etc; WES will pick up translocations involving coding regions - e.g., MYB-NFIB, MECT1-MAML2

We thank the reviewer for highlighting the importance of MYB-translocation status in ACC. We have reanalyzed the cohort following the reviewer's suggestions. We have described our findings from PLAG1- and EWSR1-fusions above. However, translocation of MYB or MYBL1 were identified in 69% of samples. No other translocation events were detected in the fusion-negative ACC. However, MYB gene upregulation was consistently identified among ACC samples (we also included these data in Figure 2), thus supporting the diagnosis. These findings suggest different mechanisms of MYB-activation but are consistent with the diagnosis of ACC.

"PC1, which separated ACC from other entities, was related to immune response (Figure S1A)." Please provide a figure within this sup figure of the PC1 loadings (what genes specifically drive this difference).

We thank the reviewer for the comment. We restructured and improved Suppl. Figure 3 to include the requested plots.

"A primary analysis of TIM composition was performed using scores for 2 different sets of signatures of immune infiltrates." Contextualize how these indices were selected? Provide references or how the authors derived this choice.

We thank the review for this comment. We have revised methods, results and figures to further contextualize the use of immune scores. Immune signatures used in this paper were selected based on several factors. They were required to be related to tumor immunity, cited in several publications and of similar sizes. The total number of genes covered by the signatures is 62 and the overlap among them was quite small with no gene present in all 6 signatures (max. jaccard's coefficient is 0.2). The only signature with no unique genes was the IFNG signature with only 6 genes in total. We updated the methods section on "Bulk RNAseq downstream analysis".

"Yet, we identified a small subset of ACC samples (n=10, ~17%) with high immune infiltration, which could also be validated in a separate analysis in only ACC samples (n=57)." This requires further analysis using the data the authors already have generated. The authors shall consider the correlation of translocation status with TMB, after assigning driver mutations amongst the available cases (as requested above). In other studies, TMB is higher in non-translocation positive tumors (e.g., salivary duct carcinoma, epithelial-myoeptithelial carcinoma with HGT). Do these high mutation burden samples lack translocations? Alternately, are these high inflammation also HGT samples? These details all require further reporting.

We agree, that molecular characteristics of this immune-high ACC subset is of interest. We have specifically analyzed translocation status, TMB and mutational

profiles in these samples but were not able to identify an association between molecular features and inflammation status. However, TMB and inflammation was higher in non-translocation positive tumors. We have added these findings to the manuscript, Figure 2 and in Suppl. figure 5.

Tumor mutational burden description - "Mutational signatures 48 were extracted for WGS data. Four main signatures could be found in 52 patients with WGS including APOBEC-signatures (SBS2,13), chemotherapy." Where is this data presented in the manuscript? mutational burden signature enrichment in each case should be included in a table linked to the anonymous identifiers used in the patient-level sup table 1. Report known driver mutations in all samples c/w TCGA top.

We agree, that mutational signatures warrant further exploration and have performed additional analyses. The results have been added to Suppl. Figure 2 and are discussed in manuscript results and discussion section. We have added the TMB data to Figure 2 and Suppl. Table 1.

"Since TMB was not associated with inflammation in ACC samples..." Where is this data or show the relationship amongst the two.

These data are now provided in Supplemental Figures 5 and 6.

Single nuclei sequencing:

The cohort is very small and heterogenous to the point that meaningful associations and conclusions cannot be drawn outside of the 5 cases of AdCC. While 5 AdCC cases were sequenced (this is in my understanding the first published scRNAseq of salivary gland tumors), how many of these cases had high and low inflammation burden? An expanded cohort of snRNAseq of additional cases or using FFPE blocks using the 10X Flex kit would vastly improve the generalizability of this data. Please explain how it is possible that not a single normal epithelial cell (duct, acini?!) is represented across 85,000 cells from primary tumors from the glands WHEN the parent block has advancing tumor - tumor stroma - parenchymal interface. Supplemental data or supplemental methods should show how/where nuclei were selected from parent frozen blocks for single nuclei. MOREOVER, showing the expression of target genes in the translocation positive cells along with VCTN1 would be interesting since it is well appreciated in the biphasic tumor that the translocation can be positive in both luminal and abluminal cells.

We thank the reviewer for these helpful comments. We agree that our single-cell cohort is too small to draw definitive conclusions regarding SGC subtypes and have revised the manuscript to emphasize this limitation. However, we believe that the overall dataset is sufficient to inform bulk-based validation results in a larger cohort and allow for an in-depth analysis of specific findings.

Following the reviewer's suggestion, we have added the requested information on inflammation status to the table summarizing single-nuclei sequencing results (Table 2).

To address the issue of normal epithelial cell representation, we conducted several analyses to determine whether any normal salivary gland epithelial cells were present in our dataset. First, we isolated all cells that did not cluster into the

malignant cluster, identifying 132 cells from a primary sample. However, these cells did not express canonical salivary gland markers but rather showed expression patterns indicative of neuronal and glial cell lineages. Second, we specifically analyzed all epithelial cells derived from primary tumor samples and found that all exhibited malignant characteristics, with no evidence of healthy salivary gland epithelial cells. Furthermore, when we integrated our dataset with reference healthy salivary gland data, we observed that some malignant cells clustered together with normal epithelial cells. However, these cells continued to express malignant markers, suggesting that while they share transcriptional similarities with normal epithelial cells, they remain cancerous. Third we looked for expression of known genes in salivary mucus and serous glandular cells. These genes were expressed at a higher level in a very small amount of cells overall, which clustered together with malignant cells.

Detecting normal epithelial cells from tumor biopsies in single-cell RNA sequencing is an ongoing challenge, not only in salivary gland studies but across various tissues. Identifying copy number variations (CNVs) and single nucleotide variations (SNVs) in single-cell RNA-seq data is difficult due to low coverage, and normal and tumor cells often have very similar transcriptomic profiles, making differentiation challenging. Additionally, salivary gland carcinomas (SGCs) are known to have low tumor mutation burden (TMB) and low CNV burden, which further complicates this task.

While we cannot rule out the presence of normal salivary gland epithelial cells in our dataset, if present, they are likely very few and difficult to identify. We acknowledge this limitation and suggest that future studies with larger cohorts and advanced sequencing technologies may improve the detection and characterization of these cells.

Following the reviewer's suggestion, we added MYB-expression status in correlation to VTCN1 expression to Figure 7.

Please separate IgG and IgA plasma cells into two clusters by manual annotation.

We added a new plot to Figure 4 which shows the expression of plasma cell markers.

Figure 3A requires case codes for each stacked bar chart.

Case codes are now found on both stacked bar charts on Figure 4. The case codes are the same as in Supplementary table 1.

Fig3C. Show M1 and M2 since you discuss M2 polarization. Show dot-plots showing expression differences and markers for M1 and M2 in the figure.

We added a plot showing requested markers for Macrophage polarization to Suppl. Figure 8 .

Please explain how immune low, medium, and high cutoff were established and how a

"low vs "high" might change the findings? It seems that medium and high almost intersect in your data whereas low seems to be distinct.

We agree, that the identification of immune subgroups is challenging. We performed supervised clustering, which revealed that three subgroups give the best representation of phenotypes (see Figure 3 and Suppl. Figure 4), even though there is an overlap . However, the overlap would be even greater with two subgroups. We have added a discussion of these considerations to the respective result section in the manuscript.

Figure 4E needs annotation (dotted outline) of the tumor or use of another color. Representative LOW and HIGH CD68 infiltration is needed. M2 vs M1 marker to confirm M2 polarization. Mucicarmine or PAS to evaluate if they are mucin laden.

We have added requested figures to Suppl. Figure 10. PAS staining was performed to rule out muciphages (as described above).

Figure 4E would be aided by IF/IHC showing CD4/CD8 T cells and/or CD38/CD138+ Plasma cells in the tumor stroma.

We have adjusted the depiction of exemplary IHC stainings according to the reviewer's suggestion.

Figure 4D should add "immune medium line" since it is reported in the preceding figures.

We agree that this is misleading. Survival plot of Figure 4D is not based on the three identified immune clusters but rather on samples with the highest and lowest percentage of T-cells (first and fourth quartile) to highlight the difference in survival better. We have highlighted this in the image caption (Figure 5) to prevent misunderstanding.

"TIM of advanced SGC to be Macrophage-dominated" The authors do not provide a definition of what an advanced SGC is versus any other carcinoma. There is no patient level data table that references this data.

We thank the reviewer for pointing out this uncertainty. The here analyzed samples are all coming from patients with recurrent and/or metastatic SGC (lack of curative treatment options is an inclusion criterion for the MASTER program), thus representing an advanced SGC cohort. We have added these considerations to the manuscript and have also added patient-level data.

"On the other hand, no significant effect of macrophage proportion was found on overall survival, although a tendency to shorter survival in samples with high macrophage content was observed (data not shown)." This data is important and consistent with other data in the manuscript. It is inappropriate to withhold it. Please include.

We agree, that it is imperative to provide representation of the data and have added the requested plot to Suppl. Figure 10

"Abundance of macrophages and T-cells were correlated inversely both in bulk data

(rho=-0.57,p=5e-09^{***}) as well as in single nuclei data (rho=-0.91,p=1e-05^{***}). Mapping of M-phage and T cell cases within one tumor type stratified by tumor stage or high grade features, or something is needed here. Please show evidence that this relates to tumor parameters/tumor type/tumor grade, etc.

T-cells and Macrophages were inversely correlated in both ACC and non-ACC at a very similar magnitude. Also this correlation was not dependent on any other parameters such as cell of origin, site of biopsy, cohort or prior therapy. However we could not test this in single cell data because of the very low number of samples available per group. We eventually decided to remove this paragraph, because additional correlations could not be identified and also decided, that an anticorrelation was as expected, thus not worthwhile reporting.

Also, the majority of macrophages were predicted as being M2 polarized (median 81% of all macrophages). Both points indicate an immune-suppressive role of macrophages in the TIM." While this is well-understood for other cancers, the authors overlook an important aspect of the salivary glands. M2 macrophages in the salivary glands cluster with the predominant macrophages in normal resident immune infiltrates in the salivary glands - so-called "muciphages", mucin-laden macrophages that engulf spilled mucins in the extracellular space. Mucin spillage is common at the advancing edge of tumors in the salivary glands given compression and disruption of the ductal network, whether or not they are immune suppressive. Some comparison to normal age-matched salivary glands' M2 macrophage phenotypes is needed to make this claim above and to show how they are phenotypically different AND immune suppressive in the SGC tumor context. An alternate explanation is that, the M2 may not be immune suppressive as stated, but merely a consequence of the tumor on the parent organ causing mucin spillage. This may include adjacent normal nuclei from FFPE blocks of the tumors for snRNAseq and IF/ISH mapping of M2 m-phage at the tumor-parenchymal interface.

We thank the reviewer for these important insights. As stated above, we did rule out the presence of muciphages immunohistochemically, following the reviewer's suggestion. In order to better characterize the physiological immune microenvironment, we integrated an analysis of healthy salivary glands (bulk and single cell), which showed distinct immune cell subsets. The overall macrophage abundance was much smaller but also showed predominant M2 polarization in bulk data. However, single-cell analyses revealed distinct macrophage phenotypes in healthy salivary glands.

Figure 5: What is the top red/mauve/pink column at the tops of the stacked bar chart? These types of omitted details are concerning.

We changed the figure and added a description of the bar-chart.

Figure 5C shows the immune clustering clusters more naturally into two clusters, not three. This could be discussed further as previously identified.

We agree, that the overlaps are more pronounced in the clustering of immune checkpoints (now Figure 7). We believe that this is linked to a more

histology-dependent expression of immune checkpoints. We have added a discussion of the number of immune clusters to the results section.

Report TMB for the ICI-treated responders/non-responders. My hypothesis is the adenocarcinoma NOS (not really an appropriate molecularly-based diagnosis without exhaustive molecular investigation showing that it is NOT a known salivary gland malignancy - especially in the face where the authors can report the mutation status of this case based on available "multiomics" data). It is curious that the two tumors responding to ICI are classes enriched for high mutational burden (Salivary duct carcinoma and high grade adenocarcinomas that are difficult to classify without appropriate ancillary testing). This is an important caveat to report.

We agree, that response to immune checkpoint inhibition is multifactorial. We also agree that adenocarcinoma NOS is not an adequate diagnosis in SGC. Following the reviewer's suggestion, we have additionally characterized the responder, which revealed an EWSR fusion, in line with the potential diagnosis of clear cell carcinoma. We have added these information, together with TMB to figure 6. We have also highlighted the limitations and multifactorial cause of immune checkpoint inhibitor response to the manuscript.

Where is the mention on what the two groups of ACC tumors are i.e., ACC1 and ACC2 groups?

We have added a definition of previously defined molecular subtypes of ACC1 and ACC2 in the methods section "Bulk RNAseq downstream analysis"

Discussion:

"Inflammation was associated with tumor mutational burden mostly in non-ACC rather than ACC histologies in our analyses, albeit the effects were marginal which might be related to an overall lower tumor mutational burden in the majority of samples" but why - please elaborate.

Salivary gland cancers with a median of 1.1 mut/MB have a rather low TMB compared to other cancer types. Looking at the whole cohort we can observe a weak correlation between the inflammation and TMB, which however disappears when stratified by ACC vs non-ACC. The few highly inflamed ACC samples also had a very low TMB indicating that inflammation must be driven by other factors, which are not caught by an analysis of the genomic landscape. We have added these considerations to the manuscript discussion section.

"The high macrophage content, in particular of M2-polarized macrophages in advanced salivary gland cancers, might therefore mediate immune checkpoint inhibitor resistance in these patients despite the presence of an inflamed TIM." Some experimental validation of this in AdCC would help make this point better.

We have deleted this sentence in the revised version in order to moderate claims on the clinical impact of these findings. We agree, that an experimental validation of findings could help to further elucidate the impact of the TIM on immune checkpoint inhibitor response. Currently, the slow growth, low take rate of ACC organoids and challenges in experimental design of preclinical immunotherapy studies prevent an

adequate preclinical validation. We are, however, working on this but believe that this is beyond the scope of the current manuscript.

Point-by-point reply to reviewer's comments.

Reviewer #1 (Remarks to the Author):

The authors have effectively addressed our concerns by collecting and validating data across multiple levels of omics. However, the T-cell/macrophage ratio remains a point of concern as a potential biomarker for clinical benefit. While they noted that clinical benefits were observed only in patients with a high T-cell/macrophage ratio, the limited number of such patients and the ambiguity surrounding the criteria for defining this ratio raise important questions. For instance, it appears that there are more patients who, despite having a T-cell/macrophage ratio similar to those who respond, do not exhibit any clinical responses. Therefore, uncertainty remains about whether a high T-cell proportion can reliably serve as a biomarker for immunotherapy. Please elaborate on this concern.

We thank Reviewer #1 for favorably assessing the revised version. We agree, that the limited number of patients does not allow for a direct clinical implementation of T-cell/macrophage ratio as a biomarker. We have therefore added the need for additional verifying data in the abstract and discussion section. We agree, that response to immune checkpoint inhibitors is likely mediated by several factors in addition to the tumor immune microenvironment, thus mediating different responses even among patients with a higher T-cell/macrophage ratio. However, we believe that the T-cell/macrophage ratio is a promising additional biomarker in these hard-to-treat tumors that warrants additional investigation.

Reviewer #2 (Remarks to the Author):

No further Comments.

Reviewer #3 (Remarks to the Author):

The authors have addressed very nearly all the reviewers' comments. Their revised manuscript has improved the manuscript in multiple ways. Most importantly, the logic of the manuscript is greatly improved.

One persistent issue is the lack of consistent and systematic organization of the figures. While a minor issue scientifically, it is disruptive to the interpretation within the scientific context of the manuscript (requires going back/forth b/w legends and text and both forwards and backwards within the figures). Some of the figures are not arranged alphabetically from left to right (e.g. B, then A). Within the same critique and similar to reviewers' comments in the previous version, the use of one legend for multiple different figures is challenging to visually (in an individual who does not have color blindness) link the legend to the figures.

Notably, the writing of the manuscript is also tremendously improved. However, there is a lack of coherence in writing between the new and the original sections. During editing, the authors may want to align the writing across the manuscript.

We thank Reviewer #3 for the favorable assessment. Following the reviewer's suggestion, we have revised all figures and arranged them alphabetically to improve readability.

We have additionally revised the manuscript text to improve writing.